# Scaling Turbulent Combustion Fields in Explosions

**Allen Kuhl [1,\*], David Grote [1] and John Bell [2,\*]**

[1]   Lawrence Livermore National Laboratory, Livermore, CA 94550, USA; grote1@llnl.gov
[2]   Lawrence Berkeley National Laboratory, Berkeley, CA 94720, USA
\*   Correspondence: kuhl2@llnl.gov (A.K.); jbbell@lbl.gov (J.B.); Tel.: +1-(925)-922-3774 (A.K.)



**Featured Application: Scaling of Explosions.**

**Abstract:** We considered the topic of explosions from spherical high-explosive (HE) charges. We studied how the turbulent combustion fields scale. On the basis of theories of dimensional analysis by Bridgman and similarity theories of Sedov and Barenblatt, we found that all fields scaled with the *explosion length scale* $r_0$. This included the blast wave, the *mean* and root mean squared (*RMS*) profiles of thermodynamic variables, combustion variables, velocities, vorticity, and turbulent Reynolds stresses. This was a consequence of the formulation of the problem and our numerical method, which both satisfied the similarity conditions of Sedov. We performed numerical simulations of 1 g charges and 1 kg charges; the solutions were identical (within roundoff error) when plotted in scaled variables. We also explored scaling laws related to three-phase pyrotechnic explosions. We show that although the scaling formally broke down, the fireball still essentially scaled with the explosion length scale $r_0$. However, the discrete Lagrange particles (DLP) (phase 2) and the heterogeneous continuum model (HCM) of the DLP wakes (phase 3) did not scale with $r_0$, and *mean* and *RMS* profiles could differ by a factor of 10 in some regions. This was because the DLP particles and wakes introduced an additional scale that broke the similarity conditions.

**Keywords:** gas dynamics; dimensional analysis; similitude theory; *mean* and *RMS* turbulent velocity profiles; turbulent Reynolds stresses; three-phase models of pyrotechnic explosions

## 1. Introduction

Similarity theory dates back to Bridgman's work [1] summarized in *Dimensional Analysis*, which he published in 1922. He points out that the differential equations describing physical phenomena can be rescaled into dimensionless units, resulting in equations that are invariant over dimensionless groups. Solutions are then invariant in settings where the dimensionless groups remain constant. This is codified in the "Buckingham $\Pi$ Theorem" [2]. Sedov [3] applied such concepts to mechanics in *Similarity and Dimensional Methods in Mechanics* (1959). His work codified and summarized similarity methods in fluid mechanics. The similarity theory was used to derive the similarity solutions of the point explosion problem first published by G. I. Taylor [4] in 1941 and later by Sedov [5] in 1946. This same similarity solution method was used by Oppenheim and Kuhl to compute all possible similarity solutions bounded by a strong shock [6], a strong detonation wave [7], and self-similar explosion waves of variable energy at the front [8]. Scaling aspects of blast waves were summarized by G. I. Barenblatt in his book *Scaling, Self-Similarity, and Intermediate Asymptotics* [9] published in 1996.

Here, we consider the topic of explosions from spherical high-explosive (HE) charges. We are interested in the turbulent combustion fields in their fireballs, as illustrated in Figure 1. In this work, we derive scaling laws for all fields (including the thermodynamic variables, velocities, component mass-fractions, and turbulent Reynolds stresses) on the basis of the similarity theory for explosion fields, as described by Sedov.

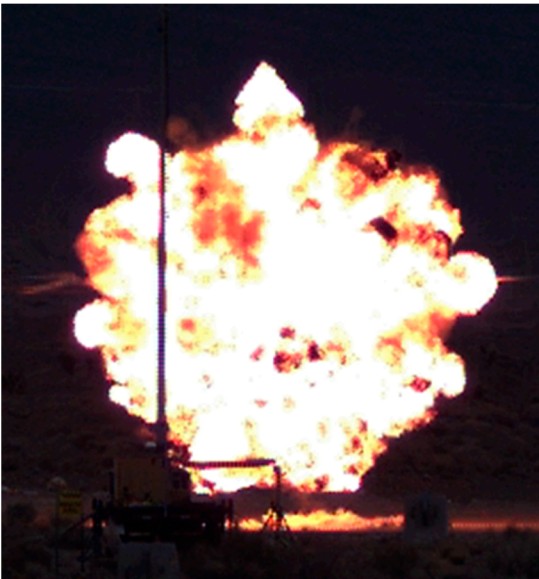

**Figure 1.** Fireball image from a 15-kg COMP B spherical charge at 4 m height of burst.

*Relevance of This Work*

One of the reasons that such scaling laws are important is because we can study the characteristics of turbulent fireballs at the laboratory scale, such as 50 g charge explosions, as reported by Glumac and Kuhl [10], then scale them up to large scale explosions, such as the 15 kg charge explosion shown in Figure 1. Many more diagnostic techniques can be applied at the laboratory scale but are impractical at the field-test scale. Such laboratory measurements include schlieren/shadow photography; holographic interferometry of particles; spectral imaging in the IR, visible, and UV wavelengths; measurements of species composition via gas chronometry; collection of carbon particles that are the source of strong Planckian radiation [10]; and optical measurements *inside the fireball,* as demonstrated in [10], to name just a few. In other words, using the scaling laws demonstrated in this paper, we can measure intimate details of turbulent fireballs in laboratory experiments, then confidently apply them to large-scale explosion events like that shown in Figure 1.

In contrast, the particle and mist phases of pyrotechnic explosions do not obey Sedov's scaling laws. Thus, calculations and experiments must be performed at each size of pyrotechnic charge under consideration. Even if the initial geometric charge size is scaled, the evolving flow fields of the particles and mist do not scale. Such facts have not been recognized in the past.

## 2. Gas Dynamics of Explosions

The turbulent combustion fields are governed by the inviscid conservation laws of gas dynamics. These may be expressed in strong conservation form as

*Mass:*

$$\partial_t \rho + \nabla \cdot \rho \boldsymbol{u} = 0 \tag{1}$$

*Momentum:*

$$\partial_t \rho \mathbf{u} + \nabla \cdot (\rho \boldsymbol{u} \boldsymbol{u} + p) = 0 \tag{2}$$

*Energy:*

$$\partial_t \rho E + \nabla \cdot (\rho E u + p u) = 0 \tag{3}$$

*Component DP:*

$$\partial_t \rho Y_D + \nabla \cdot (\rho Y_D u) = 0 \tag{4}$$

In the equations above, $\rho$ is density, $\boldsymbol{u}$ represents velocity, $E$ denotes total energy ($E = e + 0.5\,u\cdot u$), and $Y_D$ is the mass fraction of detonation product species. Two components are recognized: detonation products (DP), which serve as the fuel, and air (A). Their mass fractions obey the conservation relation:

$$Y_D + Y_A = 1 \tag{5}$$

In pure computational cells, we have either $Y_D = 1$ or $Y_A = 1$; in mixed cells, $Y_D$ and $Y_A < 1$. We assume that the components are in a state of thermodynamic equilibrium. This allows us to compute the pressure and temperature equation of states (EOS) according to

*Pressure:*

$$\mathrm{p} = f_1(\varrho, e,\, Y_D) \tag{6}$$

*Temperature:*

$$\mathrm{T} = f_2(\varrho, e,\, Y_D) \tag{7}$$

These functions are evaluated by using the thermodynamic equilibrium code Cheetah [11,12]. Results have been tabulated for computational efficiency.

We follow the evolution of component mass fractions of *fuel (detonation products):* $Y_F$, *oxidizer; (air):* $Y_O$; and *products (combustion products):* $Y_P$. They were computed according to the following conservation laws:

*Fuel ( $Y_F$):*

$$\partial_t \varrho Y_F + \nabla\cdot U \varrho Y_F = -\dot{\varphi} \tag{8}$$

*Oxidizer ($Y_O$):*

$$\partial_t \varrho Y_O + \nabla\cdot U \varrho Y_O = -\alpha\dot{\varphi} \tag{9}$$

*Products ($Y_P$):*

$$\partial_t \varrho Y_P + \nabla\cdot U \varrho Y_P = (1+\alpha)\dot{\varphi} \tag{10}$$

*Conservation:*

$$Y_F + Y_O = Y_P \tag{11}$$

where $\alpha$ denotes the stoichiometric oxidizer/fuel ratio. We use the fast chemistry, infinite Damkohler number limit (i.e., gas dynamic limit), where turbulent mixing within the cell occurs according to the MILES concept of Boris [13]:

*Combustion Rate:*

$$\dot{\varphi} = \begin{cases} \text{burn all Air}:\ \Delta\varrho_A/\varrho\Delta t & (\text{Fuel}-\text{rich cell}) \\ \text{burn all Fuel}:\ \Delta\varrho_F/\varrho\Delta t & (\text{Air}-\text{rich cell}) \end{cases} \tag{12}$$

The problem is initialized by a similarity solution of Kuhl [14] for a constant velocity detonation wave propagating at the Chapman–Jouguet (CJ) speed when the detonation wave reaches the surface of the charge. See Figure 2 for the similarity profiles. (We used a perturbation on the density profile to break nonphysical numerical symmetries. We discuss HE density perturbation effects on the *mean* and root mean squared (*RMS*) profiles in the Section 6.)

The detonation products expand, doing $p\cdot dV$ work on the surroundings, thereby creating a blast wave. The *DP*–air interface is unstable; perturbations grow due to vorticity creation by the inviscid baroclinic mechanism: $\dot{\omega} = \frac{1}{\rho^2}\nabla\rho \times \nabla p$, thereby creating a turbulent combustion layer near the edge of the fireball.

We solved the above conservation laws with an unsplit second-order Godunov method. We used the unsplit piecewise parabolic method (PPM) of Colella and Woodward [16,17] to advance the solution in time. PPM flattens slopes in cells near discontinuities to enforce monotonicity constraints (i.e., suppression of oscillations near discontinuities). This slope flattening induces a non-linear dissipation mechanism that acts on the cell level. It also reduces the scheme from second-order accurate

in smooth regions of the flow, to first order near discontinuities. This is consistent with the MILES (monotone integrated large eddy simulations) concept of Jay Boris [13], whereby mixing/dissipation only occur on the smallest grid scale. We used adaptive mesh refinement (AMR) [18,19] to follow steep gradients and mixing structures.

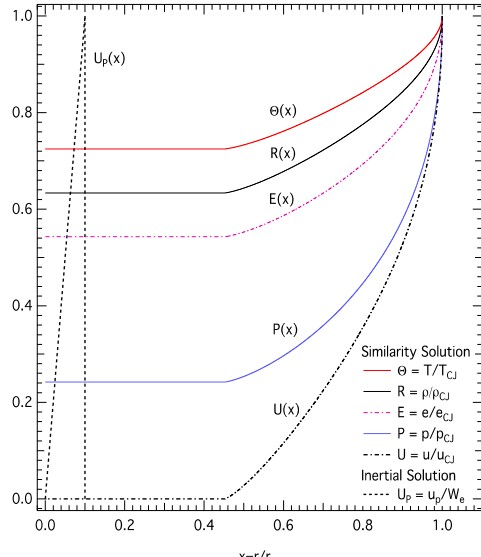

**Figure 2.** Initial conditions based on the similarity solutions for a Chapman–Jouguet (CJ) detonation wave in the gas phase by Kuhl [14], and the linear velocity profile of an inertial flow expansion into a vacuum by Stanyukovich [15] for the droplet phase.

## 3. Scaling Laws

In the inviscid gas dynamics formulation described above, variables such as density, velocity, pressure, and internal energy are related in ways controlled by their dimensional units. Sedov [3] and Barenblatt [9] show that chemical explosion fields are controlled by the non-dimensional independent variables of space, $x$, and time, $\tau$, according to

$$x = r/r_0 \tag{13}$$

$$\tau = t/t_0 \tag{14}$$

where the explosion length scale is

$$r_0 = (m \cdot \Delta H_d / p_a)^{1/(j+1)} \tag{15}$$

and the explosion time scale is

$$t_0 = r_0 / a_a \tag{16}$$

In the above, $m$ is the mass of the charge, $\Delta H_d$ represents the heat of detonation of the explosive material, and $a_a$ denotes the sound speed of the ambient atmosphere. The variable $j$ equals 0, 1, or 2 for planarly, cylindrically, or spherically symmetric flows. Similitude theory states [3] that the non-dimensional dependent variables of $U = \{\varrho, u, e, p, T\}$ of gas dynamics are functions of $x$ and $\tau$:

$$R = \varrho / \varrho_a = f_1(x, \tau) \tag{17}$$

$$R = \varrho / \varrho_a = f_1(x, \tau) \tag{18}$$

$$\varepsilon = e / e_a = f_3(x, \tau) \tag{19}$$

$$P = p/p_a = f_4(x, \tau) \tag{20}$$

$$\Theta = T/T_a = f_5(x, \tau) \tag{21}$$

For spherical explosions in three-dimensional Cartesian co-ordinates, Equation (13) becomes

$$x = x/r_0 \text{ and } y = y/r_0 \text{ and } z = z/r_0 \tag{22}$$

and the spherical ($j$ = 2) explosion length scale becomes

$$r_0 = (m \cdot \Delta H_d / p_a)^{1/3} \tag{23}$$

The dependent variables then become

$$R = \varrho/\varrho_a = f_1(x, y, z, \tau) \tag{24}$$

$$U = u/a_a = f_2(x, y, z, \tau) \tag{25}$$

$$E = e/e_a = f_3(x, y, z, \tau) \tag{26}$$

$$P = p/p_a = f_4(x, y, z, \tau) \tag{27}$$

$$\Theta = T/T_a = f_6(x, y, z, \tau) \tag{28}$$

The above is the form of the solution of the inviscid conservation laws of gas dynamics for spherical explosions. Everything scales with the *explosion length scale*, $r_0$.

## 4. Results

### 4.1. Fireball Cross-Sections

Figure 3 depicts a cross-section of the temperature field of a fireball from a 1 g TNT (Trinitrotoluene explosive: $C_6H_2(NO_2)_3CH_3$) explosion in air at 100 μs/g$^{1/3}$. One can see a variety of scales in the mixing layer. Combustion in this inviscid formulation occurs in an exothermic sheet. Our numerical method satisfies same type of scaling laws as the fireball. Consequently, we expect the numerics to also be self-similar. We confirmed this observation by simulations at two scales.

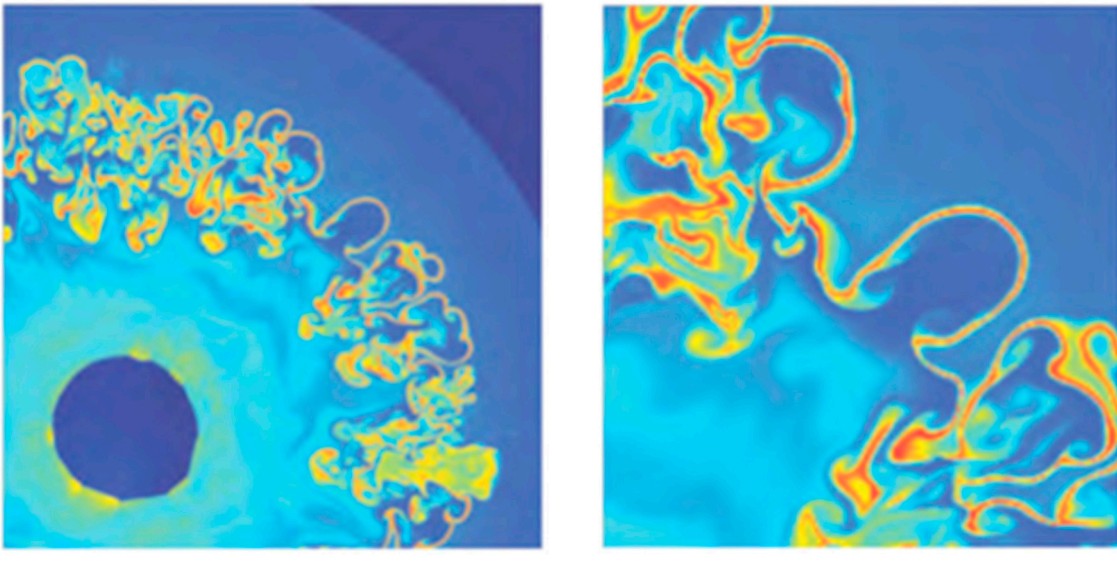

(**a**) Temperature cross-section  (**b**) Blow-up near the front

**Figure 3.** Exothermic flame sheet in the TNT–air combustion field at 100 μs/g$^{1/3}$.

Figures 4 and 5 compare cross-sections of the flow fields from a 1 g charge at $t = 0.5015$ ms/g$^{1/3}$ with flow fields from the 1 kg charge at the same scaled time, $t = 5.015$ ms/kg$^{1/3}$. Figure 4 compares the density fields; it shows the outer shock (blast wave) and secondary expanding shock, as well as the turbulent mixing in the fireball. Figure 5a,b shows comparisons of the vorticity fields in the fireballs, showing mean fireball radii of $R_{FB} = 30$ cm/g$^{1/3}$ and $R_{FB} = 300$cm/kg$^{1/3}$, respectively. These are driven by the gas dynamic fields, which scale with the explosion length scale, $r_0$.

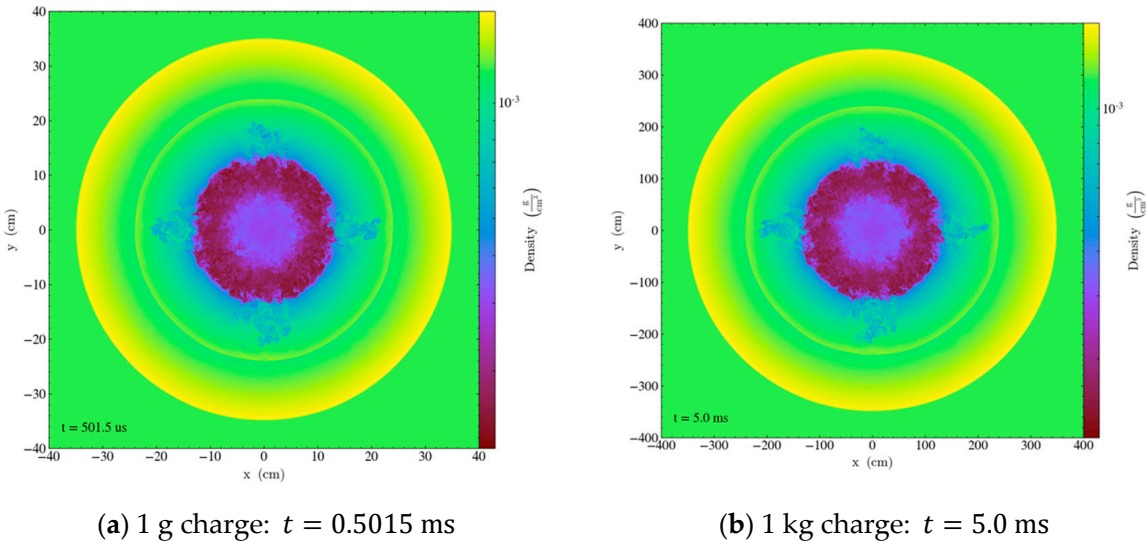

(**a**) 1 g charge: $t = 0.5015$ ms　　　　　　　　　　　(**b**) 1 kg charge: $t = 5.0$ ms

**Figure 4.** Comparison of the cross-section of the density fields.

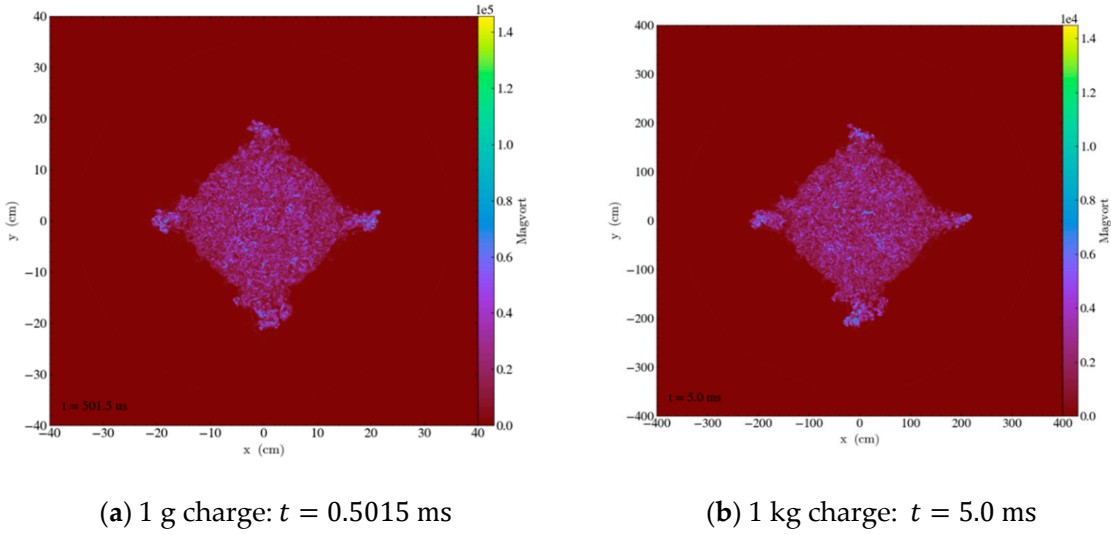

(**a**) 1 g charge: $t = 0.5015$ ms　　　　　　　　　　　(**b**) 1 kg charge: $t = 5.0$ ms

**Figure 5.** Comparison of the cross-section of the vorticity fields.

It is difficult to decern differences in the two flow fields from the above cross-section figures. Thus, we computed the maximum density difference between the two cases at each scaled time. Results are shown in Figure 6. Differences started off as roundoff errors $O \sim (10^{-13})$. They grew exponentially at around 5 μs/g$^{1/3}$, and eventually saturated at values of tens of percent.

### 4.2. Mean and RMS Profiles

To quantitatively compare the flow field differences in Figures 4 and 5, we azimuthally averaged the three dimensions fields in $\theta$ *and* $\phi$ to produce *mean* and *RMS* (root mean squared) radial profiles of the turbulent variables. The averaging method is presented in Appendix A.

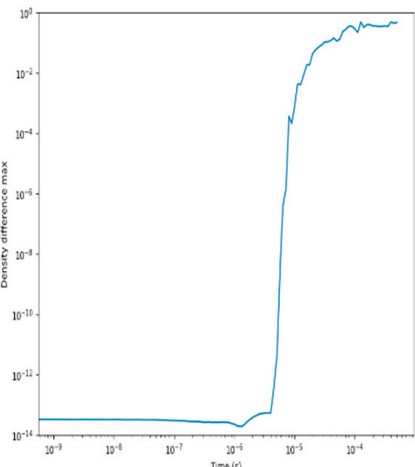

**Figure 6.** Maximum density difference between the 1 g case and the 1 kg case as a function of time. Differences stayed at roundoff errors until 5 µs, when they grew exponentially at late times.

Figures 7 and 8 depict azimuthally averaged profiles of the thermodynamic profiles of temperature, pressure, and density from a 1 g charge at $t = 0.5015$ ms (blue curves) compared with flow fields from the 1 kg charge at the same scaled time, $t = 5.015$ ms (orange curves). The *mean* and *RMS* profiles from the two scales overlayed.

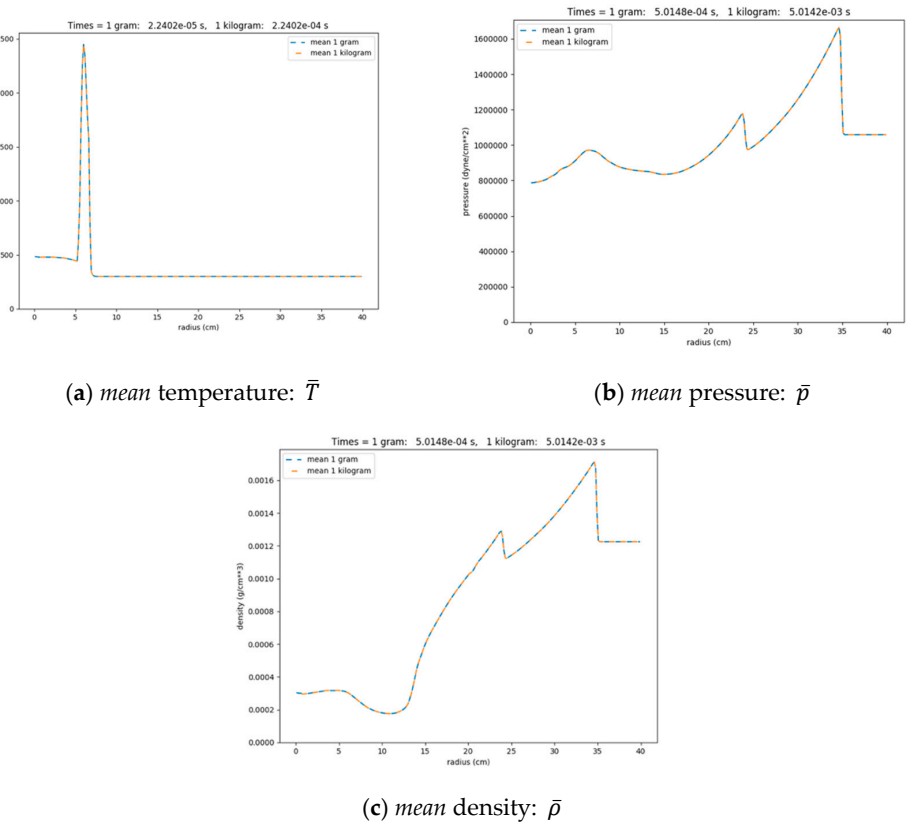

(**a**) *mean* temperature: $\overline{T}$

(**b**) *mean* pressure: $\overline{p}$

(**c**) *mean* density: $\overline{\rho}$

**Figure 7.** Comparison of the *mean* thermodynamic profiles at $t = 0.5015$ ms (1 g charge) and $t = 5.0$ ms (1 kg charge).

Figures 9 and 10 present the azimuthally averaged combustion profiles of fuel (detonation products), oxidizer (air), and combustion products from the 1 g charge at $t = 0.5015$ *ms* (blue curves)

compared with flow fields from the 1 kg charge at the same scaled time, $t = 5.015$ ms (orange curves). The *mean* and *RMS* profiles from the two scales overlayed.

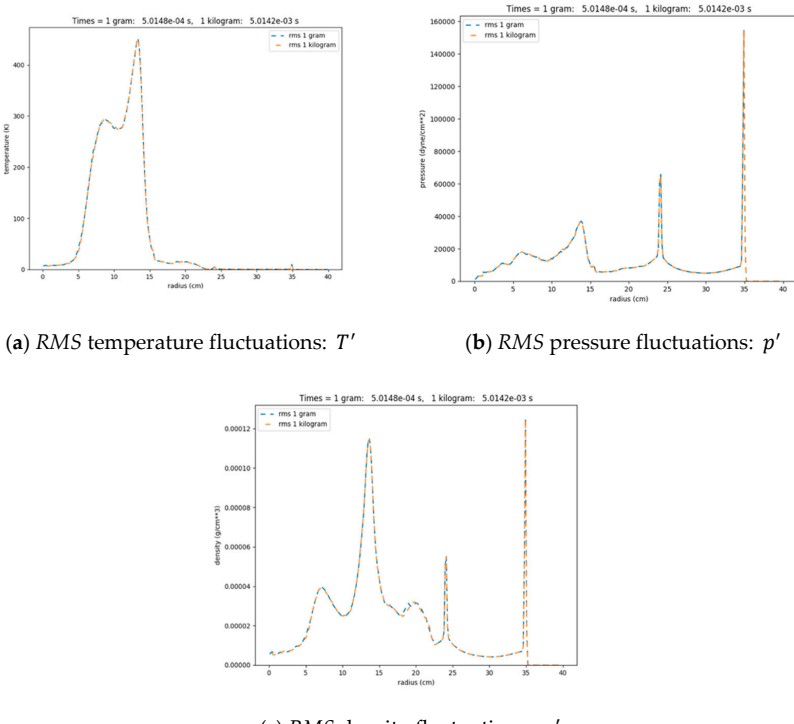

(**a**) *RMS* temperature fluctuations: $T'$

(**b**) *RMS* pressure fluctuations: $p'$

(**c**) *RMS* density fluctuations: $\rho'$

**Figure 8.** Comparison of the *RMS* thermodynamic profiles at $t = 0.5015$ ms (1 g charge) and $t = 5.0$ ms (1 kg charge).

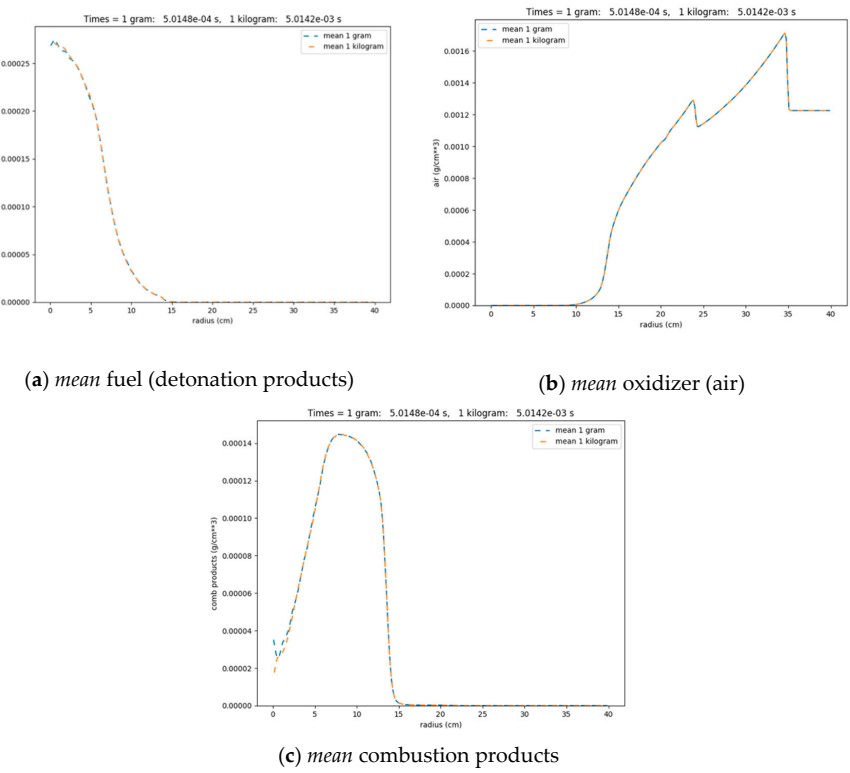

(**a**) *mean* fuel (detonation products)

(**b**) *mean* oxidizer (air)

(**c**) *mean* combustion products

**Figure 9.** Comparison of the *mean* combustion component profiles at $t = 0.5015$ ms (1 g charge) and $t = 5.0$ ms (1 kg charge).

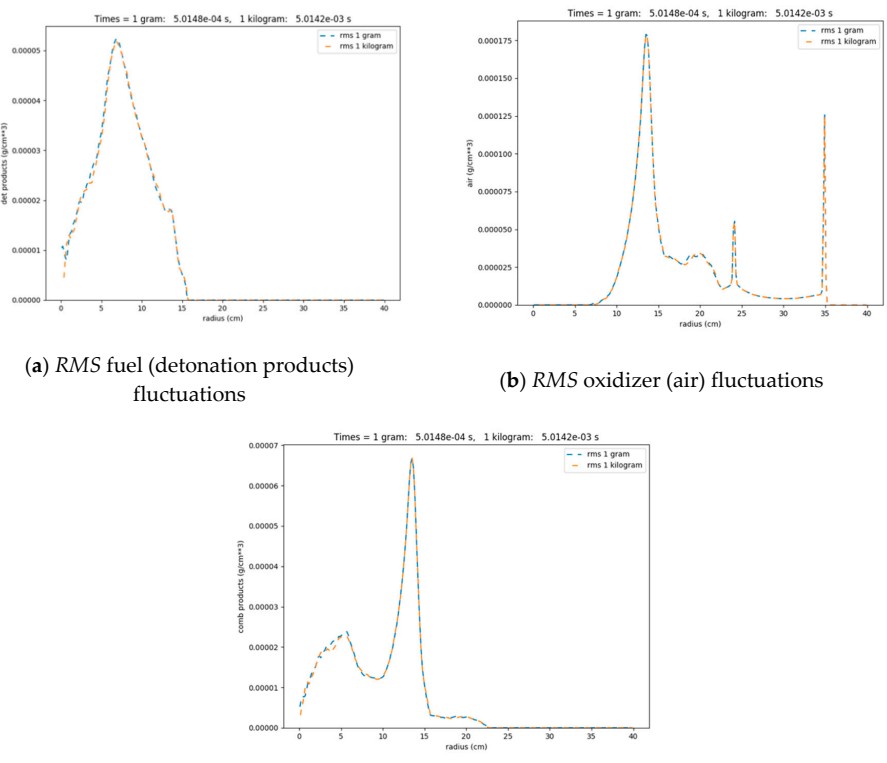

(**a**) *RMS* fuel (detonation products) fluctuations

(**b**) *RMS* oxidizer (air) fluctuations

(**c**) *RMS* combustion product fluctuations

**Figure 10.** Comparison of the *RMS* combustion component profiles at $t = 0.5015$ ms (1 g charge) and $t = 5.0$ ms (1 kg charge).

Figures 11 and 12 depict azimuthally averaged velocity profiles ($u_r, u_\theta,$ and $u_\phi$) from a 1 g charge at $t = 0.5015$ ms (blue curves) compared with flow fields from the 1 kg charge at the same scaled time, $t = 5.015$ ms (orange curves). The *mean* and *RMS* profiles from the two scales overlayed. Profiles of mean values for $u_\theta$ and $u_\phi$ were essentially zero.

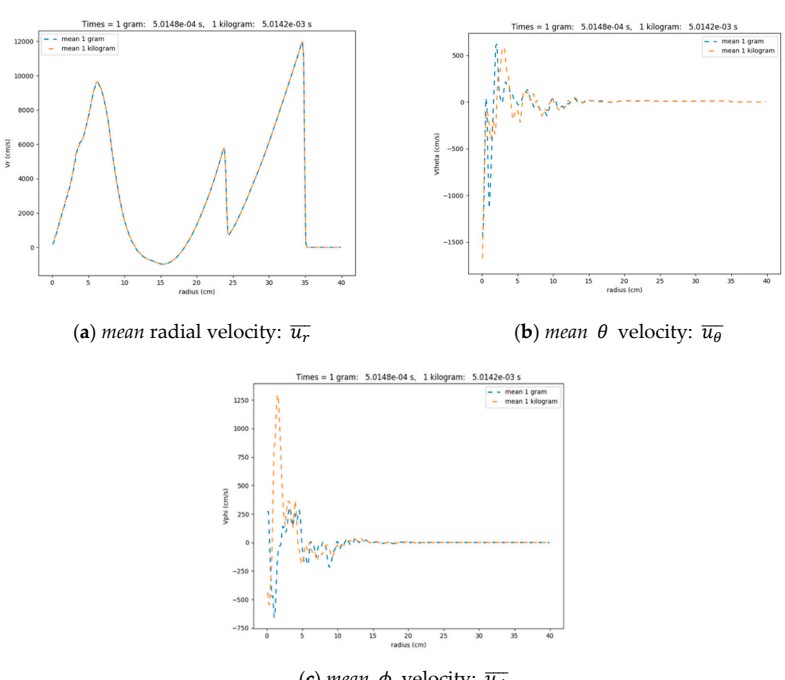

(**a**) *mean* radial velocity: $\overline{u_r}$

(**b**) *mean* $\theta$ velocity: $\overline{u_\theta}$

(**c**) *mean* $\phi$ velocity: $\overline{u_\phi}$

**Figure 11.** Comparison of the *mean* velocity profiles at $t = 0.5015$ ms (1 g charge) and $t = 5.0$ ms (1 kg charge).

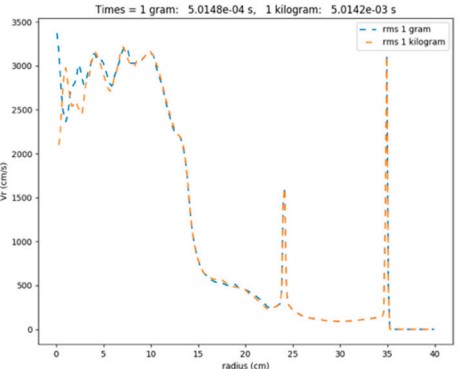

(**a**) *RMS* radial velocity fluctuations: $u'_r$

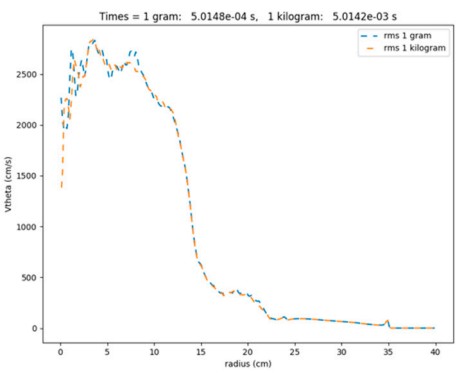

(**b**) *RMS mean* $\theta$ velocity fluctuations: $u'_\theta$

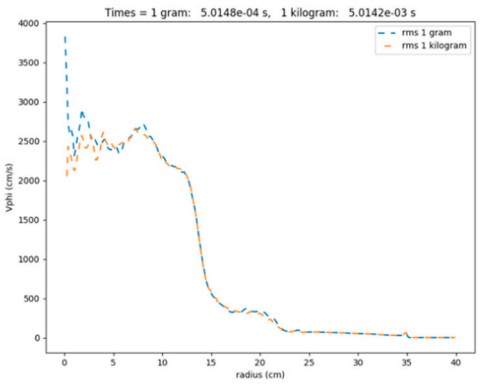

(**c**) *RMS* $\phi$ velocity fluctuations: $u'_\phi$

**Figure 12.** Comparison of the *RMS* velocity profiles at $t = 0.5015$ ms (1 g charge) and $t = 5.0$ ms (1 kg charge).

Figure 13 shows azimuthally averaged off-diagonal turbulent Reynolds stresses ($\tau_{r\theta}$, $\tau_{r\phi}$, and $\tau_{\theta\phi}$) from a 1 g charge at $t = 0.5015$ ms, compared with flow fields from the 1 kg charge at the same scaled time, $t = 5.015$ ms. The profiles were essentially zero, with random fluctuations due to the small ensemble size at small radii.

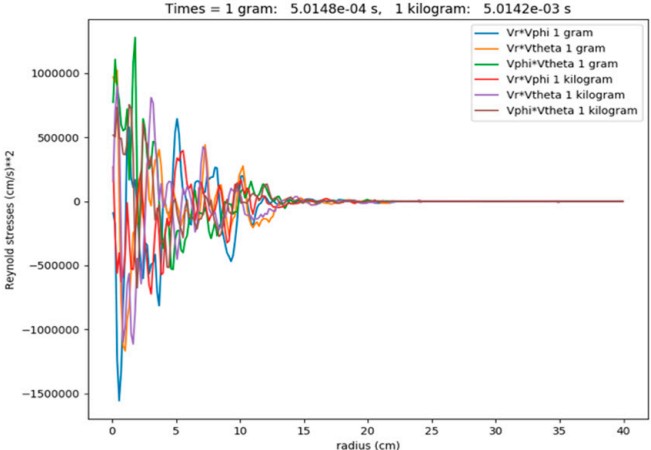

**Figure 13.** Off-diagonal turbulent Reynolds stresses were essentially zero.

Figure 14 shows the maximum density histories for a 1 g charge (blue curve) and a 1 kg charge (orange curve); the curves overlayed. Maximum densities decayed because of the adiabatic expansion of the detonation products, with a spike at $t = 0.1$ ms/g$^{1/3}$ due to the imploding shock [20].

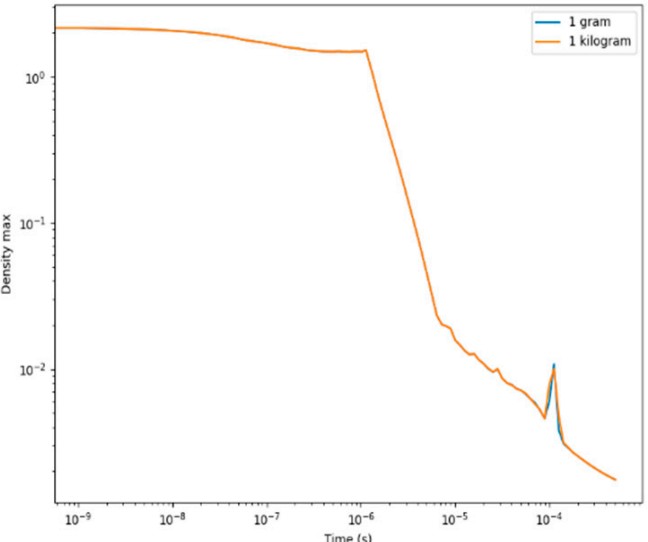

**Figure 14.** Maximum density histories for 1 g and 1 kg charges.

Figure 15 presents the combustion histories for 1 g and (blue curve) and scaled 1 kg charge (orange curve); the curves overlayed. About 50% of the fuel mass was consumed by combustion at this time.

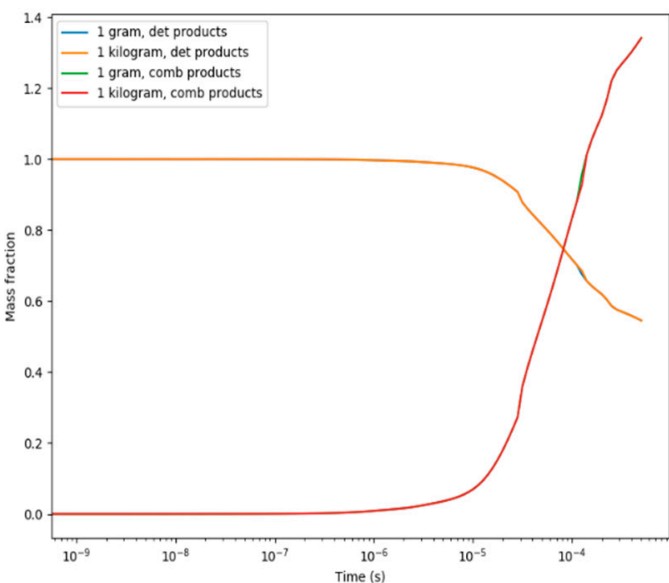

**Figure 15.** Combustion histories of fuel and products for 1 g and 1 kg charges.

Let us azimuthally average the fireball temperature field in $\theta, \phi$ to find the evolution of the mean temperature field with time:

$$\overline{T}_{\mathrm{FB}}(x, \tau) = \iint T(x, \theta, \phi, \tau) \mathrm{d}\theta \mathrm{d}\phi \tag{29}$$

Let us define the fireball radius as the radius where $\overline{T}_{FB}(x_{FB}, \tau) = T_{max}/2$ (starting from outside the fireball and working inward). Then, the evolution of the fireball radius becomes

$$x_{FB} = \widetilde{g}(\tau) \tag{30}$$

Dividing by the initial nondimensional charge radius $x_c = r_c/r_0$, one finds

$$R_{FB}/R_c = g(\tau) \tag{31}$$

From experiments and numerical simulations, when the spherical fireball has expanded to one atmosphere, one obtains

$$R_{FB}/R_c = g(\infty) \approx 30 \tag{32}$$

The largest characteristic scale in the turbulent fireball is the fireball radius, and thus turbules in the fireball scale with $r_{FB}$.

### 4.3. Summary

We confirmed the similarity of turbulent combustion fields of fireballs from HE explosions. Figure 16 presents turbulent fireball structures from our gadynamic simulations (Figure 16a,c), which are compared with a photograph of a fireball (Figure 16b). They show similar turbulent structures that scaled with the explosion length scale, $r_0$.

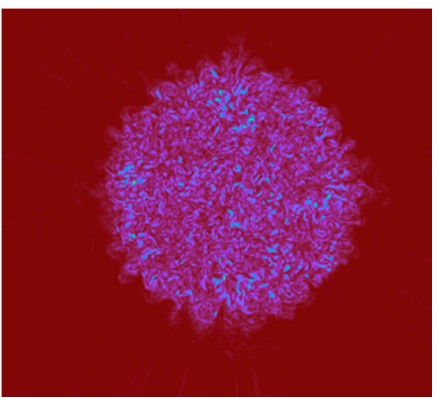

(**a**) Vorticity cross-section of a pyrotechnic explosion of 7175 kg Comp B at $t$ = 219 ms

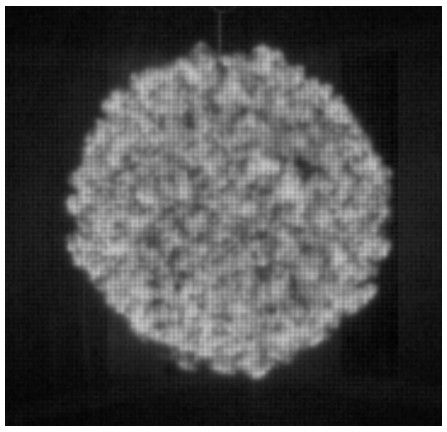

(**b**) Fireball photograph from a 50 g N5 spherical charge

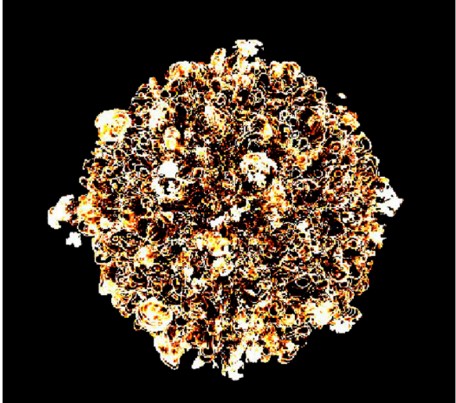

(**c**) MAUI/LUX image of a 15 kg Comp B fireball

**Figure 16.** Turbulent fireball structures from simulations (**a,c**) are compared with an experimental photograph in frame (**b**) from Glumac and Kuhl [10].

As expected, all gasdynamic profiles (e.g., thermodynamic profiles, combustion profiles, velocity profiles, and Reynolds stresses), both their *mean* and *RMS* values, scaled with the explosion scale $r_0 = (m \cdot \Delta H_d / p_a)^{1/3}$. The fireball radius scaled according to $R_{FB}/r_0 = f(t/t_0)$. To sum up, all aspects of gaseous turbulent combustion in HE explosions scaled with the explosion radius $r_0$.

## 5. Three-Phase Pyrotechnic Explosions

Next, we considered pyrotechnic explosions. Figure 17 presents a cross-section of the three-phase temperature field of a pyrotechnic explosion from our paper in the 16th International Detonation Symposium [21]. It shows the turbulent mixing and combustion of the gas phase, which reached a temperature of 2500 (yellow regions) corresponding to the adiabatic flame temperature of the HE–air system. It also shows discrete Lagrange particles (black dots) that shed micro-mist wakes (red curves). This three-phase nature of such pyrotechnic explosions is what we explore in this section. Models are described next.

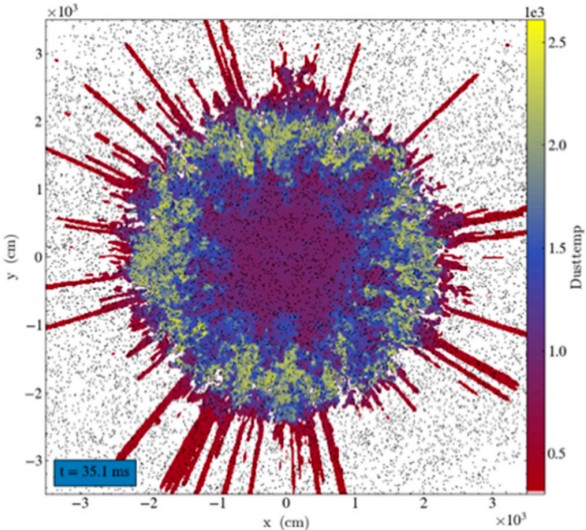

**Figure 17.** Cross-section of a pyrotechnic explosion predicted by our three-phase code [21].

### 5.1. Gas Phase

The conservation laws of gas dynamics are used to model the expansion of the detonation products, mixing, and turbulent combustion of the detonation products with air. The gas phase is coupled to the particles and wakes phases with drag and heat transfer terms. The conservation laws of mass, momentum, total energy, and detonation products (DP) mass fraction are

$$\partial_t \rho + \nabla \cdot (\rho u) = 0 \tag{33}$$

$$\partial_t \rho u + \nabla \cdot (\rho u u + p) = +D_i + n_w D_w \tag{34}$$

$$\partial_t \rho E + \nabla \cdot (\rho E u + p u) = +(D_i + n_w D_W) \cdot u + \dot{Q}_i + n_w \dot{Q}_w \tag{35}$$

$$\partial_t \rho Y_D + \nabla \cdot (\rho Y_D u) = 0 \tag{36}$$

where $\rho$, $u$, $p$, $T$, and $Y_D$ are the gas density, velocity, pressure, temperature, and mass-fraction of DP, respectively. Here, $E = e + u \cdot u / 2$ is the total energy and $e$ is the internal energy. The equation of state is specified by tables $p$, $T$, $a = f_i(\rho, e, Y_D)$ based upon the Cheetah code [11]. Drag and heat transfer effects with the particles and wakes are included in the source terms on the right-hand side of Equations (34) and (35).

### 5.2. Particle Phase

The dynamics of the particle phase is modeled by discrete Lagrange particles (DLPs) and their interactions with air, namely, mass, momentum, and energy [22]:

$$\frac{dx_i}{dt} = v_i(x_i) \tag{37}$$

$$\frac{dm_i}{dt} = -\dot{s}_i(x_i) \tag{38}$$

$$m_i \frac{dv_i}{dt} = -D_i(x_i) - \dot{s}_i v_i + m_i \text{g} \tag{39}$$

$$m_i \frac{de_i}{dt} = -\dot{Q}_i(x_i) - \dot{s}_i e_i \tag{40}$$

where $x_i$, $v_i$, $m_i$, and $e_i$ denote the position, velocity, mass, and internal energy of particle $i$, and g denotes gravity. Mass loss, drag, and heat transfer interactions with air are specified by the following relations:

$$\dot{s}_i = s d_i^2 \tag{41}$$

$$D_i = (1/8)\pi d_i^2 \rho(u - v_i)|u - v_i|C_D \text{ for } m_i \tag{42}$$

$$\dot{Q}_i = \pi d_i k(T_i - T) Nu \text{ for } m_i \tag{43}$$

where $k$ represents thermal conductivity [$k$] = joules/(sec-m-K). The surface recession rate, $\dot{s}_i$, is based on measurements [23] and empirical modeling [24,25] of Al–air combustion systems. The following drag and Nusselt correlations [26] (Appendix C) are assumed:

$$C_D = 0.48 + 28 Re^{-0.85} \tag{44}$$

$$Nu = 2 + 1.4 Pr^{1/3} Re^{1/5} + 0.13 Pr^{1/3} Re^{0.7} \tag{45}$$

### 5.3. Wake Phase

The wake phase is modeled by the heterogeneous continuum (HC) model of Nigmatulin [27] and Collins et al. [28]:

$$\partial_t \sigma + \nabla \cdot (\sigma v) = \dot{s}_i \tag{46}$$

$$\partial_t \sigma v + \nabla \cdot (\sigma v v) = \dot{s}_i v_i - n_w D_w \tag{47}$$

$$\partial_t \sigma E_w + \nabla \cdot (\sigma E_W v) = \dot{s}_i e_i - n_w \dot{Q}_w \tag{48}$$

Drag and heat transfer interactions between wakes and gas are modeled by

$$D_w = (1/8)\pi d_w^2 \rho(u - v)|u - v|C_D \tag{49}$$

$$\dot{Q}_w = \pi d_w k(T_w - T) Nu \tag{50}$$

$$n_w = \sigma/m_w \text{ with } m_w = (1/6)\pi d_w^3 \rho_w \tag{51}$$

The same drag and Nusselt correlations of Equations (44) and (45) are used.

### 5.4. Scaling of Pyrotechnic Explosions

The scaling laws for the gas phase were discussed in Section 2. For pyrotechnic explosions, we must deal with all three phases. One can say that the conservation laws for phases 2 and 3 describe inertial flow, since there are no pressure gradient forces. However, there are source/sink terms on the right-hand side of the equations, expressing mass transfer, drag, and heat transfer with the gas phase.

They are functions of the Reynolds and Prandtl numbers. How these scale with change in problem scale is discussed below.

The initial conditions for a pyrotechnic charge are illustrated in Figure 2. It shows the detonation wave structure for the gas phase (as before) and the linear velocity profile for the droplet phase. This corresponds to similarity solution of inertial flow expansion into a vacuum, as published by Stanyukovich [15] in 1960. The ball radius of the droplet phase extends to 10% of the charge radius (i.e., $R_B = 0.1 \cdot R_c$). These initial conditions scale with the explosion scale, $r_0$, in other words $x_B = R_B/r_0 = 0.1 \cdot R_c/r_0$. Thus, we consider two pyrotechnic explosion scales:

- A full-scale charge (the dynamics of this pyrotechnic charge system was published in the *16th Detonation Symposium* [21]) where $R_c \equiv 100$ cm with a mass of 7175 kg;
- 1/3 scale charge where $R_c \equiv 100/3$ cm with a mass of 265.7 kg.

As mentioned above [21], the pyrotechnic charge contains a liquid ball of radius $R_B = 0.1 \cdot R_c$, and thus a full-scale charge would have a ball radius of $R_B = 10$ cm, while a 1/3-scale charge would have a ball radius of $R_B = 10/3$ cm. The ball is made up of droplets; we estimated the droplet diameter as $d_i = 3$ mm, which will be the same for both scales. (We assumed the liquid sphere was put under strong radial strain, creating droplets. According to Tolman (1949) [29], the droplet size is controlled by surface tension, which is a material property. Thus, the droplet size will be the same for full scale and subscale.) The number of droplets scaled as $n_i = (2R_B/d_i)^3$, which was equal to 296,296 droplets at full scale and 10,974 at 1/3 scale. The droplet wake diameter scaled as $d_w = d_i/100$, and thus was $d_w = 3$ μm in both cases. The wake equilibration length scaled as $l_w/d_w = 10,000$, and thus the wake equilibration length was 30 cm in both cases. The droplet Reynolds number scaled as $Re_i = u \cdot d_i/v$. Assuming a characteristic velocity of 1 km/s, one finds the droplet Reynolds number of $Re_i = 2 \times 10^5$, and the wake Reynolds number of $Re_W = 2 \times 10^3$. These have drag coefficients of $C_D = 0.48$ and Nusselt numbers of $Nu \sim 40$ (see Appendix C for more Nusselt number correlations versus data). These results are summarized in Table 1. We note that here the particle sizes introduced an additional length scale, and thus we did not expect to maintain self-similarity.

**Table 1.** Comparison of radii, drag, and heat transfer for pyrotechnic droplet explosions ($d_i$ = 3 mm).

| Variable | Scaling | Scale = 1 | Scale = 1/3 |
|---|---|---|---|
| | ———*Initial Conditions*——— | | |
| HE charge radius | $R_c$ | 100 cm | 100/3 cm |
| HE charge mass (Comp B) | $M_c = \rho_0 4\pi R_c^3/3$ | ~ 7175 kg | ~ 265.7 kg |
| Liquid ball radius; diameter | $R_B = 0.1 \cdot R_c$; $D_B = 2R_B$ | 10 cm; 20 cm | 10/3; 20/3 cm |
| Droplet diameter | $d_i = 3$ mm | 3 mm | 3 mm |
| Number of droplets | $n_i = (2R_B/d_i)^3$ | 296,296 | 10,974 |
| Droplets per unit volume , $\eta_i$ | $\eta_i = 6 \cdot (2R_B/d_i)^3/\pi(2R_B)^3$ | 37/cc | 37/cc |
| Droplet equilibration length, $l_i$ | $l_i/d_i = 10,000$ | 30 m | 30 m |
| Droplet wake diameter, $d_w$ | $d_w = d_i/100$ | 3 μm | 3 μm |
| Wake equilibration length, $l_w$ | $l_w/d_w = 10,000$ | 30 cm | 30 cm |
| | ——— Drag & Heat Transfer Effects ——— | | |
| Droplet Reynolds number * | $Re_i = u \cdot d_i/v$ | $2 \times 10^5$ | $2 \times 10^5$ |
| Droplet drag coefficient | $C_D = 0.48 + 28/Re_i^{0.85}$ | 0.481 | 0.481 |
| Droplet Nusselt number [23] | $Nu = f(Re, Pr)$ | 40 | 40 |
| Wake Reynolds number | $Re_w = u \cdot d_w/v$ | $2 \times 10^3$ | $2 \times 10^3$ |
| Wake drag coefficient | $C_D = 0.48 + 28/Re_w^{0.85}$ | 0.48 | 0.48 |
| Wake Nusselt number [23] | $Nu = f(Re, Pr)$ | 40 | 40 |

* characteristic velocity $u$ = 1 km/s is assumed.

## 5.5. Gas Phase Results

We investigated the scaling behavior by comparing simulations at two different scales. Pyrotechnic fireball temperature cross-sections at 1/3 scale and full scale are presented in Figure 18.

The azimuthally averaged pyrotechnic fireball temperature profiles are presented in Figure 19. Curves of the *mean* and *RMS* profiles are shown; profiles at 1/3 scale (blue curves) and full scale (orange curves) overlayed, i.e., they scaled with $r_0$. It is interesting to note that the maximum mean fireball temperature in the pyrotechnic explosion was $\overline{T} = 1650$ K, which was lower than the HE fireball case of $\overline{T} = 2450$ K (see Figure 7a) because of heat losses to the DLPs.

Pyrotechnic fireball vorticity cross-sections are presented in Figure 20. Their azimuthally averaged *mean* and *RMS* profiles are depicted in Figure 21; results for 1/3 scale (blue curves) and full scale (orange curves) overlayed, i.e., the scale with $r_0$.

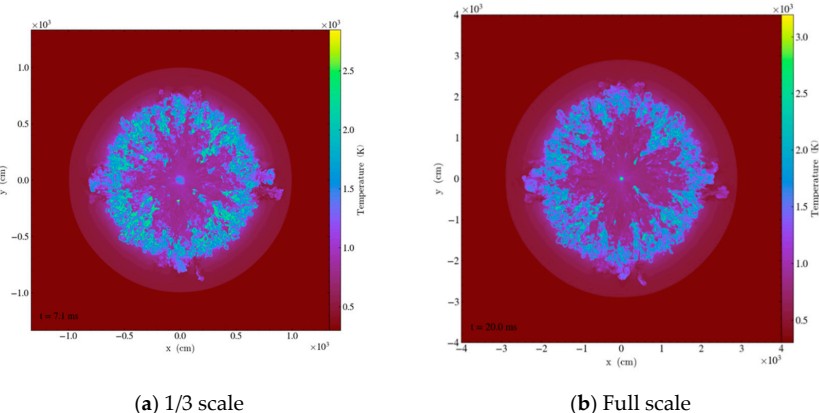

(**a**) 1/3 scale      (**b**) Full scale

**Figure 18.** Pyrotechnic fireball temperature cross-section at early times.

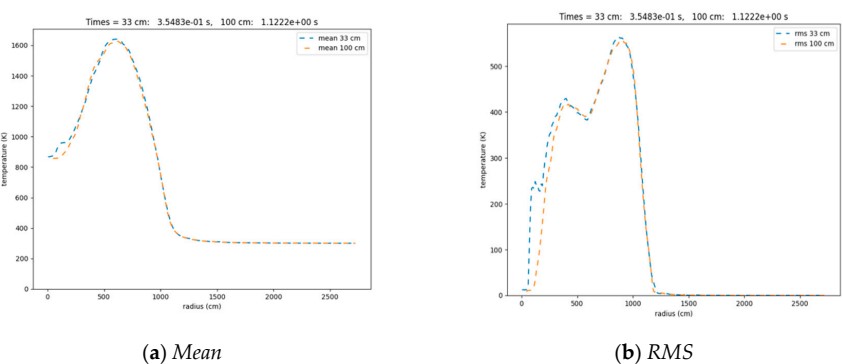

(**a**) *Mean*      (**b**) *RMS*

**Figure 19.** Azimuthally averaged pyrotechnic fireball temperature profiles.

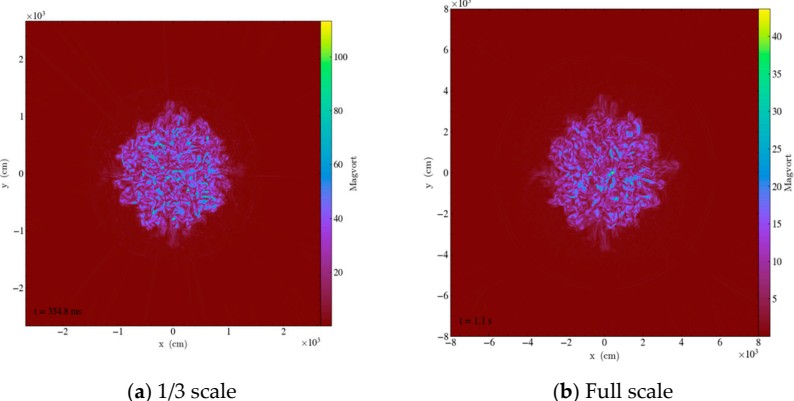

(**a**) 1/3 scale      (**b**) Full scale

**Figure 20.** Pyrotechnic fireball vorticity cross-sections.

Pyrotechnic fireball combustion product cross-sections are presented in Figures 22 and 23. Their azimuthally averaged *mean* and *RMS* combustion product profiles are shown. Results for 1/3 scale (blue curves) and full scale (orange curves) overlayed, i.e., the scale with $r_0$.

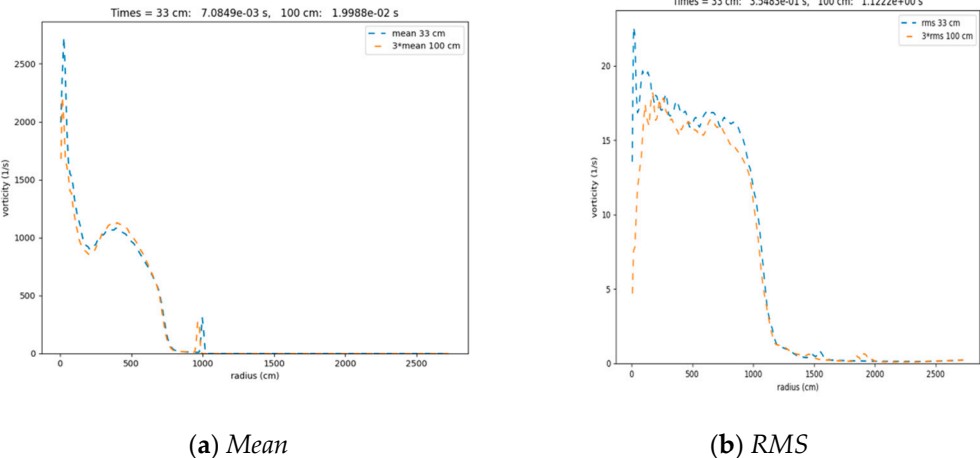

(**a**) *Mean*                    (**b**) *RMS*

**Figure 21.** Azimuthally averaged fireball vorticity profiles for pyrotechnic explosion.

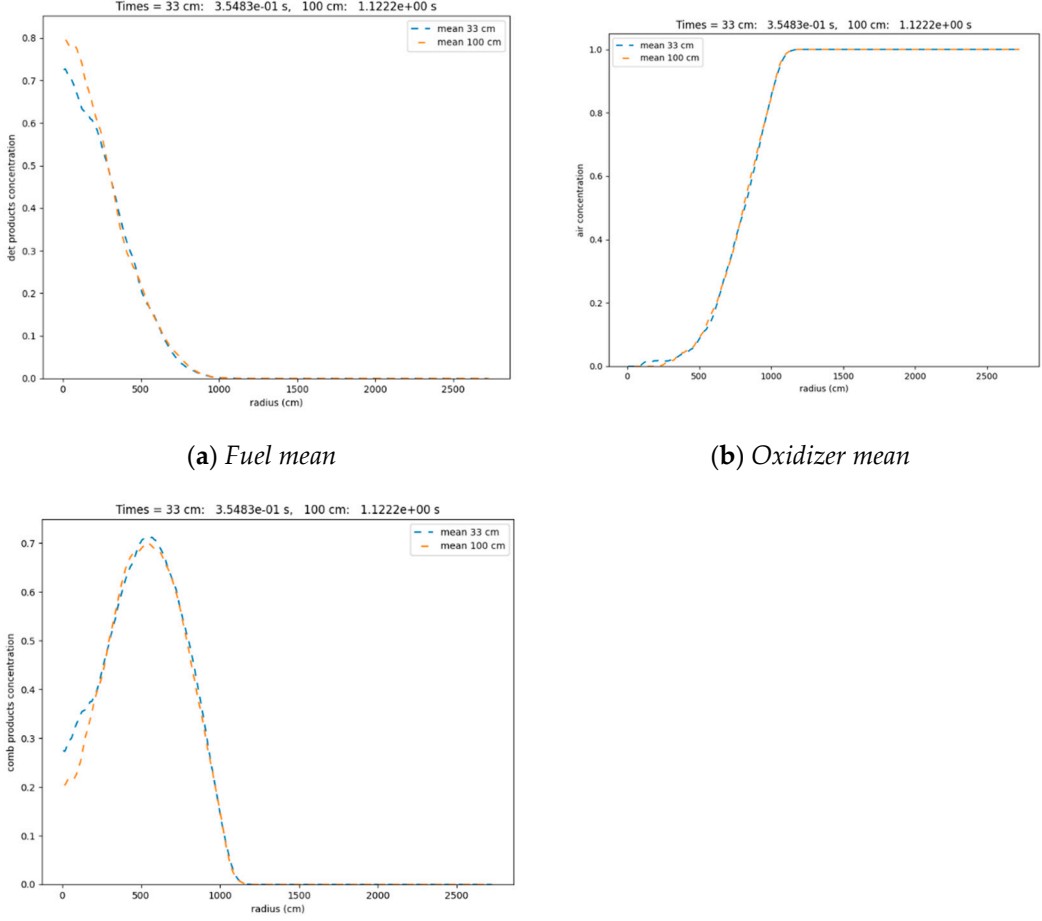

(**a**) *Fuel mean*                    (**b**) *Oxidizer mean*

(**c**) *Combustion products mean*

**Figure 22.** Azimuthally averaged *mean* combustion profiles for pyrotechnic explosions.

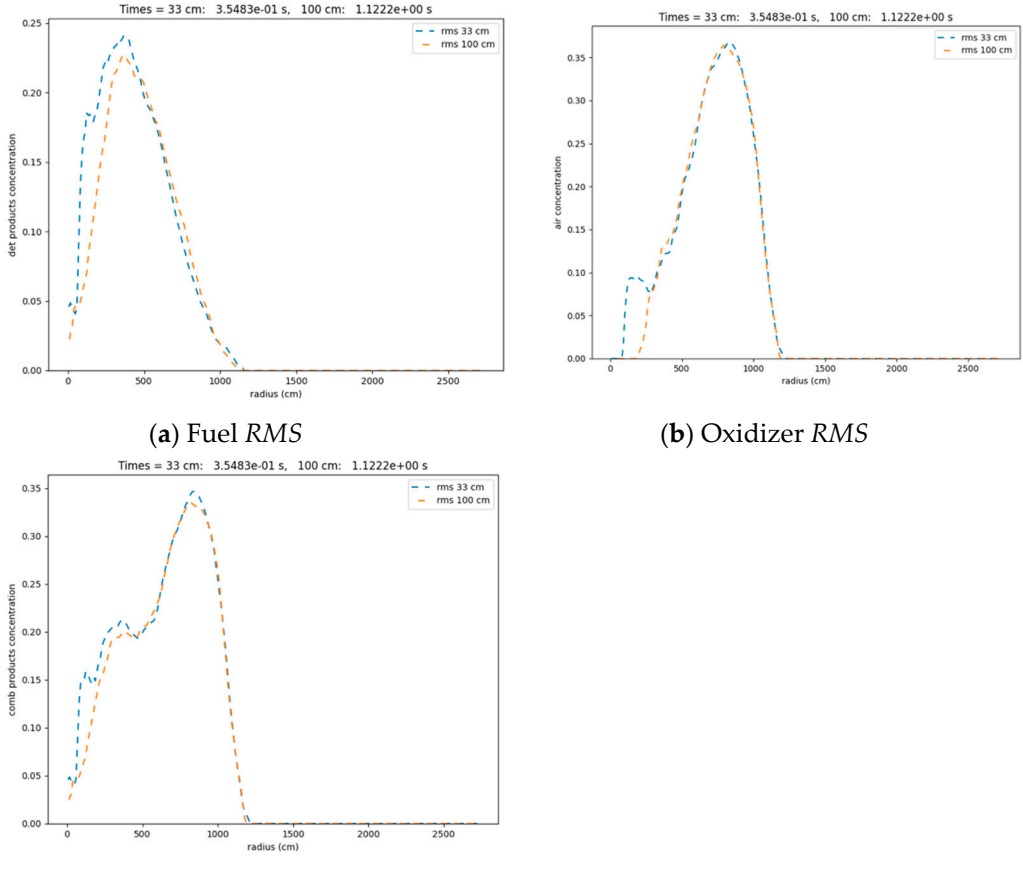

(**a**) Fuel *RMS*

(**b**) Oxidizer *RMS*

(**c**) Combustion products *RMS*

**Figure 23.** Azimuthally averaged *RMS* combustion fluctuation profiles for pyrotechnic explosions.

### 5.6. DLP Phase Velocity Results

Azimuthally averaged *mean* and *RMS* velocity profiles for the DLP phase of pyrotechnic explosions are presented in Figures 24 and 25. The *mean* and *RMS* velocity profiles of the DLP phase did not overlay and thus the DLP phase results ***did not scale with*** $r_0$.

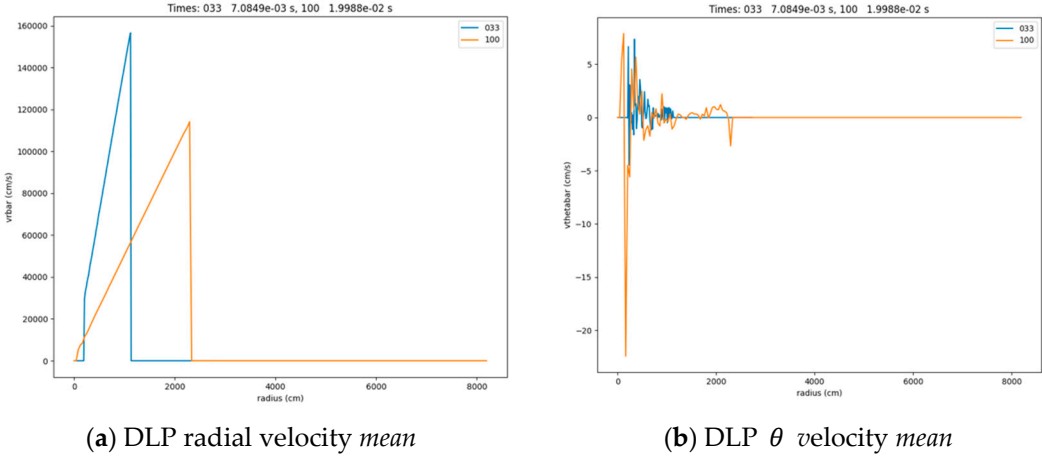

(**a**) DLP radial velocity *mean*

(**b**) DLP $\theta$ velocity *mean*

**Figure 24.** *Cont.*

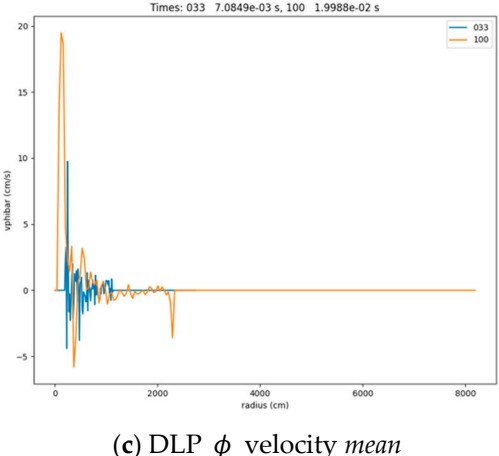

(**c**) DLP $\phi$ velocity *mean*

**Figure 24.** Azimuthally averaged mean discrete Lagrange particle (DLP) velocity profiles for pyrotechnic explosions.

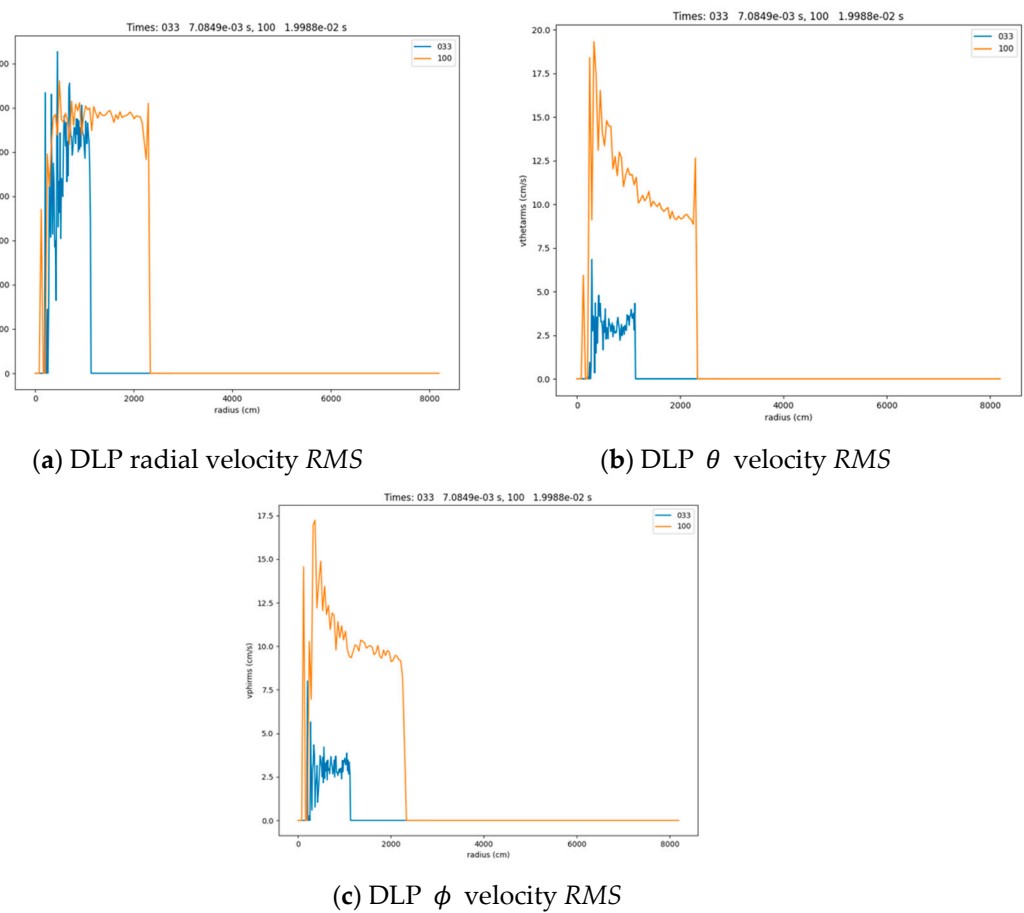

(**a**) DLP radial velocity *RMS*    (**b**) DLP $\theta$ velocity *RMS*

(**c**) DLP $\phi$ velocity *RMS*

**Figure 25.** Azimuthally averaged *RMS* DLP velocity profiles for pyrotechnic explosions.

*5.7. Mist Phase Results*

Pyrotechnic fireball mist temperature cross-sections are presented in Figure 26. Their azimuthally averaged *mean* and *RMS* mist temperature profiles are shown in Figure 27. Results for 1/3 scale (blue curves) and full scale (orange curves) did not overlay, i.e., their profiles ***did not scale with*** $r_0$.

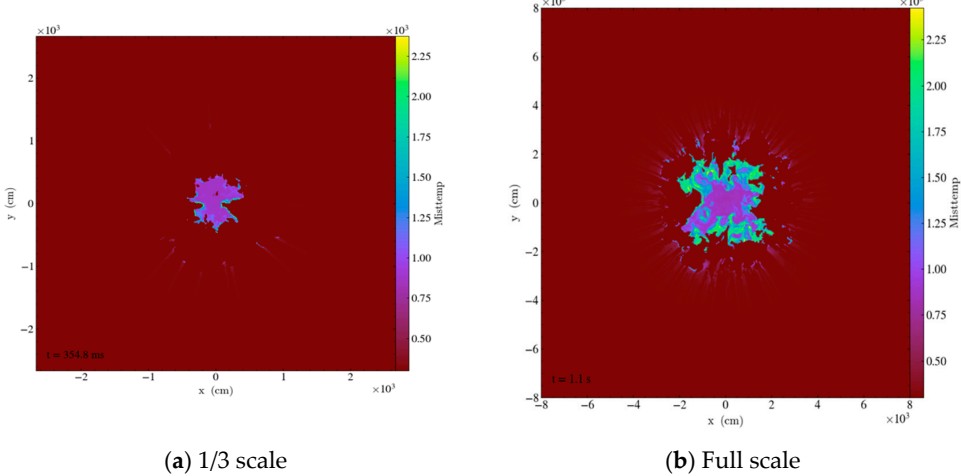

(**a**) 1/3 scale

(**b**) Full scale

**Figure 26.** Fireball mist temperature cross-sections for pyrotechnic explosions.

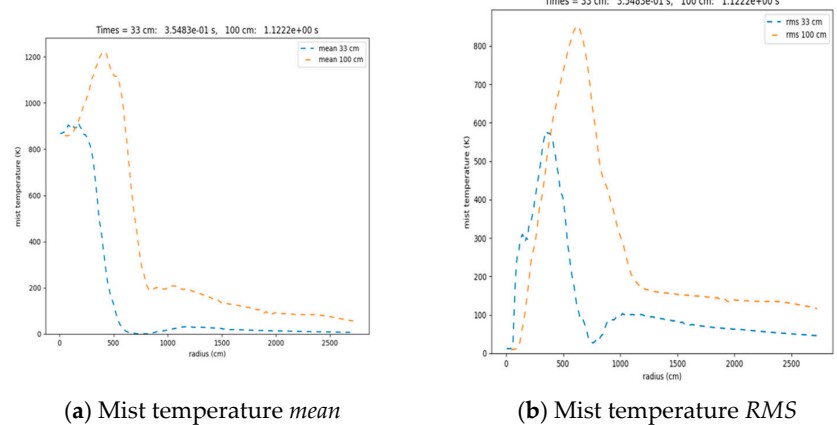

(**a**) Mist temperature *mean*

(**b**) Mist temperature *RMS*

**Figure 27.** Azimuthally averaged mist temperature profiles for pyrotechnic explosions.

Pyrotechnic fireball mist density cross-sections are presented in Figure 28. Their azimuthally averaged *mean* and *RMS* mist temperature profiles are shown in Figure 29. Results for 1/3 scale (blue curves) and full scale (orange curves) did not overlay, i.e., their profiles ***did not scale with*** $r_0$.

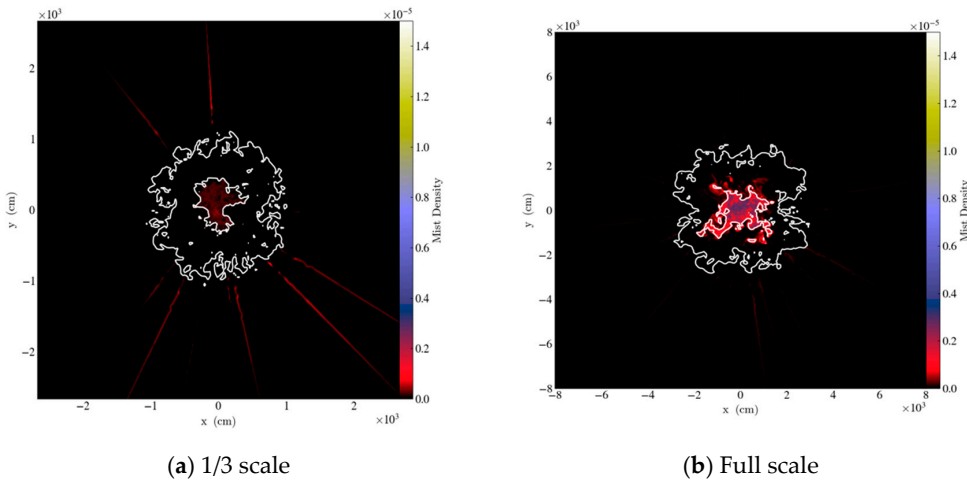

(**a**) 1/3 scale

(**b**) Full scale

**Figure 28.** Fireball mist density cross-sections for pyrotechnic explosions.

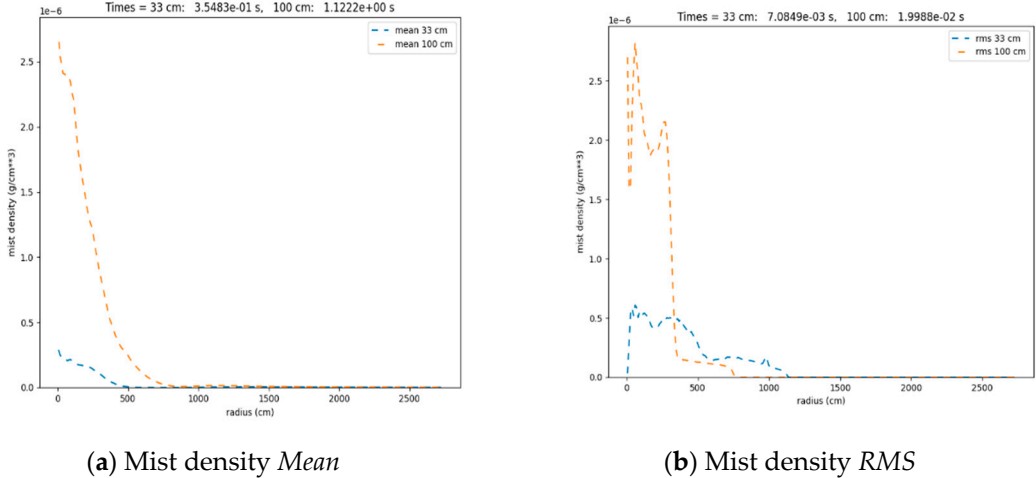

(**a**) Mist density *Mean*                    (**b**) Mist density *RMS*

**Figure 29.** Azimuthally averaged mist density profiles for pyrotechnic explosions.

The azimuthally averaged *mean* and *RMS* mist velocity profiles are shown in Figures 30 and 31. Results for 1/3 scale (blue curves) and full scale (orange curves) did not overlay, i.e., their profiles ***did not scale with*** $r_0$.

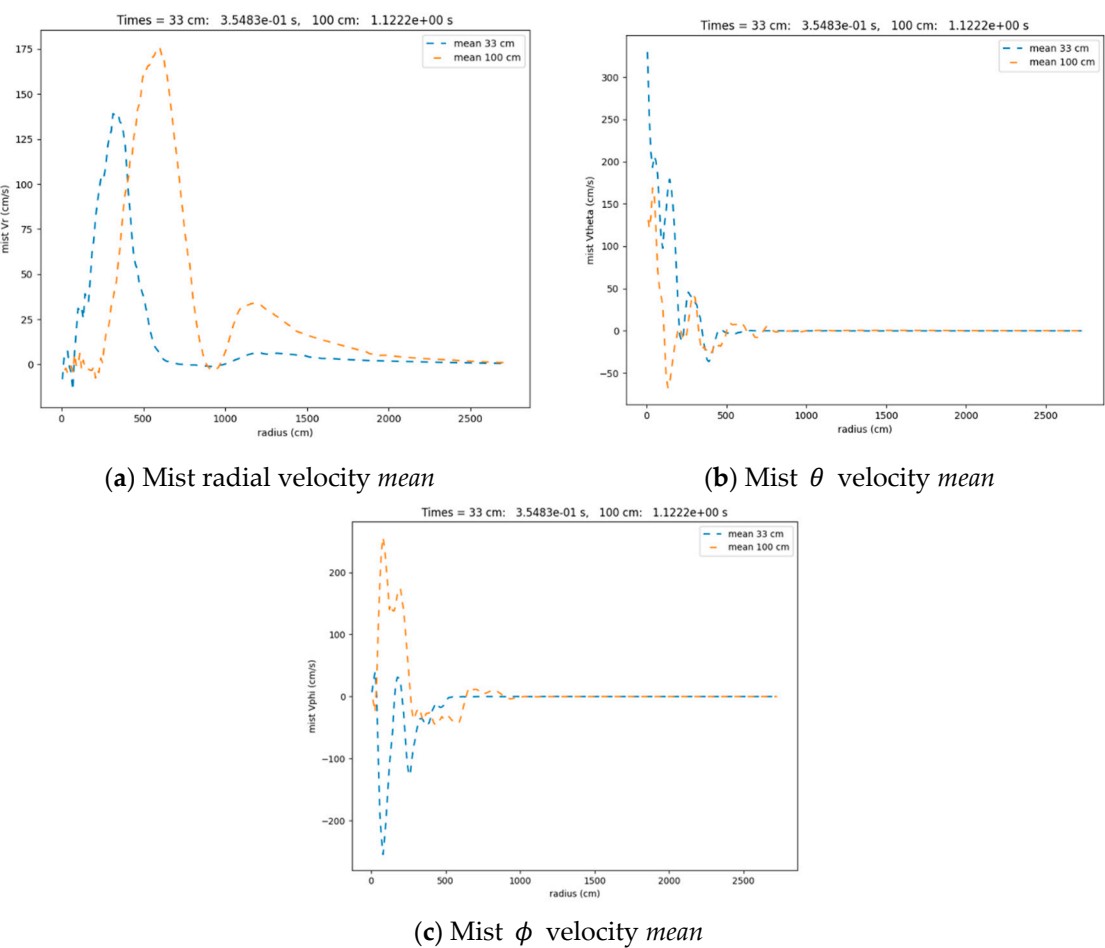

(**a**) Mist radial velocity *mean*                    (**b**) Mist $\theta$ velocity *mean*

(**c**) Mist $\phi$ velocity *mean*

**Figure 30.** Azimuthally averaged mean mist velocity profiles for pyrotechnic explosions.

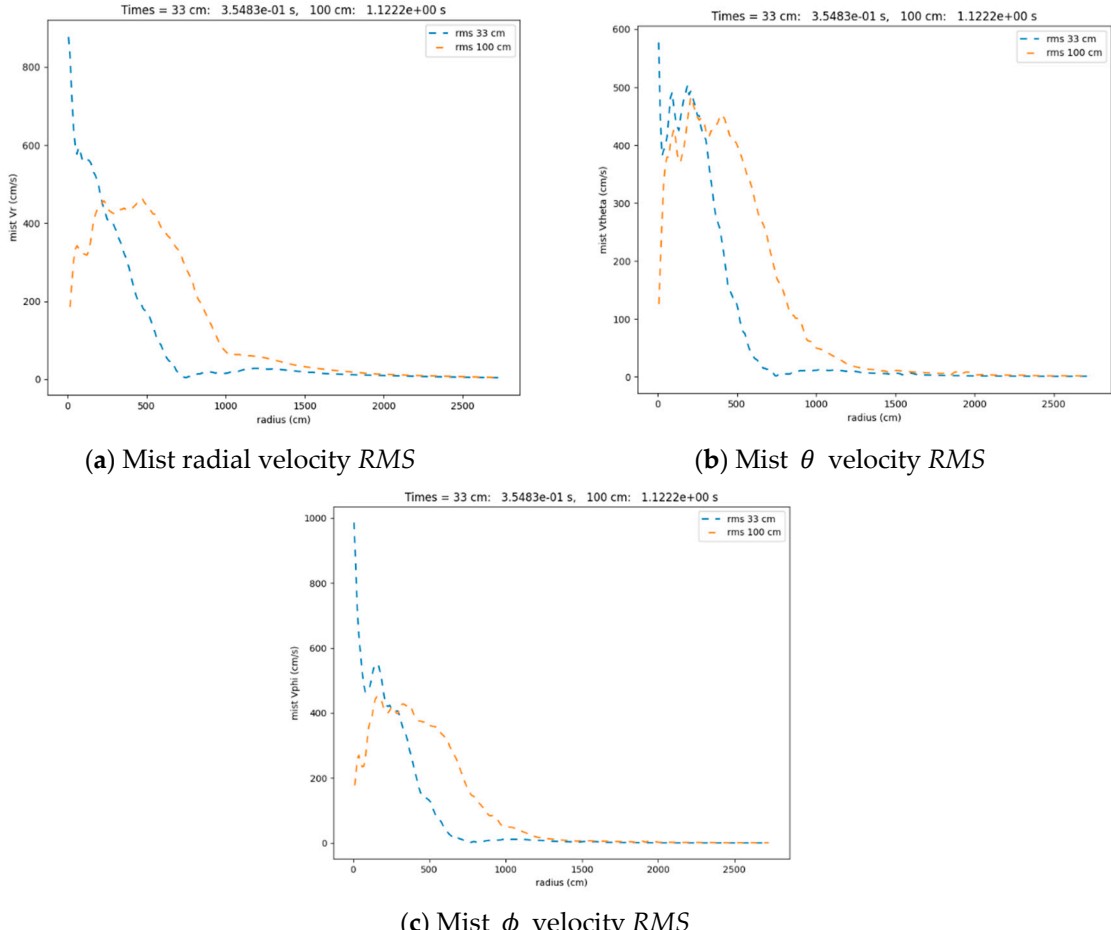

(**a**) Mist radial velocity *RMS*

(**b**) Mist $\theta$ velocity *RMS*

(**c**) Mist $\phi$ velocity *RMS*

**Figure 31.** Azimuthally averaged *RMS* mist velocity fluctuation profiles for pyrotechnic explosions.

*5.8. Summary*

A summary of the scaling laws for pyrotechnic explosions is given in Table 2. When implemented in the three-phase model (Equations (33)–(51)), one finds that

- Turbulent combustion in the gas phase fireball does scale with $r_0$;
- The DLP flow field does not scale with $r_0$;

**Table 2.** Scaling summary.

| Variable | Scaled |
|---|---|
| HE charge radius, $R_C$ | $x_C = R_C/r_0$ |
| Ball radius, $R_B$ | $x_B = R_B/r_0 = R_C/3r_0 = x_C/3$ |
| Droplet diameter, $d_i$ * | $d_i = 3 \text{ mm} = \text{constant}$ |
| Wake mist diameter, $d_w$ | $d_w = d_i/100 = 30 \text{ μm} = \text{constant}$ |
| Number droplets, $n_D$ | $n_D = (2R_B/d_i)^3$ |
| Droplets per unit volume, $\eta_i$ | $\eta_i = 6\cdot(2R_B/d_i)^3/\pi(2R_B)^3 = \text{constant}$ |
| Droplet Reynolds number, $Re_i$ | $Re_i = d_i\cdot u/v = 2\times10^5 = \text{constant}$ |
| Wake Reynolds number, $Re_w$ | $Re_w = d_w\cdot u/v = 2\times10^3 = \text{constant}$ |

* $d_i$ is controlled by surface tension; this value is estimated from experiments.

The mist flow field does not scale with $r_0$.

### 5.9. Conclusions

The gas phase of pyrotechnic explosions scaled with $r_0$, indicating that the multi-phase effects did not disrupt the scaling behavior. The DLP particle phase behaved ballistically, being only subject to drag and heat transfer without pressure (blast wave) effects, and thus did not satisfy similarity conditions. The mist phase started with zero mass and was only created by a mass sink in the DLP equations. It also was inertial (i.e., devoid of pressure effects) and only subject to drag and heat transfer effects, and thus it also did not satisfy the similitude conditions.

## 6. Discussion

### 6.1. Initial HE Density Perturbations

When initializing the HE density profile shown in Figure 2, we introduced spatial sinusoidal variations with 1% maximum amplitude to break nonphysical numerical symmetries. This helped trigger the instabilities on the HE–air interface in order to make it less dependent on grid effects. To check whether these spatial perturbations might have any effect on scaling, we changed the sinusoidal base, and re-ran the HE fireball case shown in Figures 4–15. The result was that the profiles of the two cases overlayed, and thus scaled with $r_0$.

### 6.2. Grid Refinement

We also explored the effect of grid refinement on scaling. We doubled the resolution and re-ran the HE fireball case shown in Figures 3–15. Table 3 summarizes the profile comparisons of 1× versus 2× zoning. We found that a factor of 2 increase in grid resolution had no effect on the *mean* and *RMS* profiles of thermodynamic variables, $T$, $\rho$, $m_F(t)$, and *mean* velocity-related profiles of $\omega, \mathbf{v}_r, \mathbf{v}_\theta, \mathbf{v}_\phi$. The *RMS* profiles also scaled with $r_0$, but had about 30% variation due to grid resolution.

**Table 3.** Summary of grid resolution on *mean* and *RMS* profiles.

| Variable | *mean* | *RMS* |
|:---:|:---:|:---:|
| $T$ | OK * | OK |
| $\rho$ | OK | OK |
| $m_F(t)$ | OK | – |
| $\omega$ | ~ OK ($\delta = 9\%$) | ~ OK ($\delta = 30\%$) |
| $\Delta$ | OK | OK |
| $\mathbf{v}_r$ | OK | OK |
| $\mathbf{v}_\theta$ | OK | ~ OK ($\delta = 30\%$) |
| $\mathbf{v}_\phi$ | OK | ~ OK ($\delta = 30\%$) |

* OK ≡ scales with $r_0$, $\delta$ ≡ local solution variation, resolution 1 vs. 2.

## 7. Conclusions

We investigated scaling behavior for turbulent combustion fields in explosions. Sedov has derived a similarity theory for explosions, stating that the flow fields $U(x, y, z, \tau)$ scale as $x = r/r_0$ and $\tau = t/t_0$, where $r_0 = (m \cdot \Delta H_d / p_a)^{1/(j+1)}$ and $t_0 = r_0/a_a$. This theory was derived for the blast wave (dilatational) velocity field. However, according to the Helmholtz decomposition theorem [30–32], the velocity vector field $\boldsymbol{u}$ can be divided into its dilatational $u_\Delta$ and rotational $u_\omega$ components (see Appendix B). The dilatational component corresponds to the blast wave solution of Sedov. The rotational component corresponds to the turbulent velocity field found in the fireball of explosions. Our algorithm numerically satisfies the same scaling behavior discretely, and thus we expect the numerical solution for inviscid

gaseous explosions to show the same scaling relation. The two solutions were identical when displayed in scaled variables (i.e., the flow field profiles only differed by machine roundoff).

Numerical simulations of pyrotechnic fireballs performed at two different scales confirmed this observation; their azimuthally averaged scaled profiles overlayed, proving that they scaled with $r_0$. However, the DLP flow fields and the wake flow fields of pyrotechnic explosions did not scale with $r_0$ because particle sizes introduced another length scale that broke similarity. For example, scaled mist temperature, density, and velocity profiles can differ by a factor of 10 in certain regions. Thus, calculations and numerical simulations of pyrotechnic explosions must be performed at each scale of interest. This has not been recognized in the past.

**Author Contributions:** A.K. was responsible for the gas-dynamic formulation of the three-phase models used in this research; D.G. performed the numerical simulations reported here; J.B. led the development of the high-order Godunov algorithms and Adaptive Mesh Refinement methods used in the simulations reported here. All authors have read and agreed to the published version of the manuscript.

**Funding:** Lawrence Livermore National Laboratory is operated by Lawrence Livermore National Security, LLC, for the U.S. Department of Energy, National Nuclear Security Administration under Contract DE-AC52-07NA27344. This work was funded by the Office of Defense Nuclear Nonproliferation Research and Development.

**Conflicts of Interest:** The authors declare no conflict of interest.

### Nomenclature

| Symbol | Definition |
|---|---|
| $\rho$ | gas density |
| $u$ | gas velocity vector |
| $p$ | gas pressure |
| $E$ | gas total energy: $E = e + 0.5u \cdot u$ |
| $e$ | gas internal energy |
| $Y_D$ | mass fraction detonation products DP |
| $Y_A$ | mass fraction of air |
| $Y_F$ | mass fraction of fuel (detonation products) |
| $Y_O$ | mass fraction of oxidizer (air) |
| $Y_P$ | mass fraction of combustion products |
| $\dot{\varphi}$ | combustion rate |
| $\alpha$ | stoichiometric air/fuel ratio |
| $\omega$ | vorticity vector |
| $\dot{\omega}$ | baroclinic vorticity production: $\dot{\omega} = \nabla\rho \times \nabla p / \rho^2$ |
| $x$ | nondimensional radius: $x = r/r_0$ |
| $\tau$ | nondimensional time: $\tau = t/t_0$ |
| $r_0$ | explosion length scale: $r_0 = (m \cdot \Delta H_d / p_a)^{1/(j+1)}$ |
| $t_0$ | explosion time scale: $t_0 = r_0/a_a$ |
| $m$ | mass of the high explosives charge |
| $\Delta H_d$ | heat of detonation |
| $j$ | geometry index $j = 0, 1$ or $2$ for planar, cylindrical, or spherically symmetric flow |
| $R$ | nondimensional density function: $R = \varrho/\varrho_a$ |
| $U$ | nondimensional velocity function: $U = u/a_a$ |
| $\varepsilon$ | nondimensional internal energy function: $\varepsilon = e/e_a$ |
| $P$ | nondimensional pressure function: $P = p/p_a$ |
| $\Theta$ | nondimensional temperature function: $\Theta = T/T_a$ |
| $U_\Delta$ | nondimensional dilatational velocity: $U_\Delta = u_\Delta/a_a$ |
| $U_\omega$ | nondimensional rotational velocity: $U_\omega = u_\omega/a_a$ |
| $x, y, z$ | nondimensional Cartesian coordinates: $x = x/r_0, y = y/r_0, z = z/r_0$ |
| $R_{FB}$ | fireball radius |
| $R_c$ | high explosives charge radius |
| $R_B$ | liquid ball radius |

| | |
|---|---|
| $M_c$ | high explosives charge mass |
| $n_i$ | number of $i$ droplets: $n_i = (2R_B/d_i)^3$ |
| $\eta_i$ | number of droplets $i$ per unit volume: $\eta_i = 6{\cdot}(2R_B/d_i)^3/\pi(2R_B)^3$ |
| $l_i$ | droplet $i$ equilibration length |
| *mean* | azimuthally averaged mean value |
| *RMS* | azimuthally averaged root mean squared value |
| $u_\Delta$ | dilatational velocity |
| $u_\omega$ | rotational velocity |
| $\boldsymbol{D}$ | drag force |
| $\dot{Q}$ | heat transfer |
| $n_w$ | number of particles per unit volume |
| $\boldsymbol{x_i}$ | position of particle $i$ |
| $\boldsymbol{v_i}$ | velocity of particle $i$ |
| $m_i$ | mass of particle $i$ |
| $\dot{s}_i$ | surface recession rate of particle $i$ |
| $e_i$ | specific internal energy of particle $i$ |
| $d_i$ | diameter of particle $i$ |
| $C_D$ | drag coefficient |
| $Nu$ | Nusselt coefficient |
| $Re$ | Reynolds number |
| $Pr$ | Prandtl number |
| $k$ | thermal conductivity |
| $\sigma$ | density of the wake phase |
| $\boldsymbol{v}$ | velocity of the wake phase |
| $E_w$ | energy of the wake phase |
| $n_w$ | number of wake particles per unit volume: $n_w = \sigma/m_w$ |
| $m_w$ | mass of the wake phase in cell: $m_w = (1/6)\pi d_w^3 \rho_w$ |
| $\Delta\rho_A$ | mass of air in cell |
| $\Delta\rho_F$ | mass of fuel in cell |
| Subscripts | |
| *0* | characteristic scale |
| *a* | ambient conditions |
| *FB* | fireball |
| *c* | charge |
| *i* | particle $i$ |
| *D* | drag |
| *w* | wake |
| *B* | ball |
| *r* | radial velocity |
| $\theta$ | theta velocity |
| $\phi$ | phi velocity |
| *CJ* | Chapman–Jouguet state |

## Appendix A Azimuthal Averaging

We took advantage of the spherical symmetry of the problem and azimuthally averaged the flow fields. We transformed Cartesian coordinates to spherical coordinates: $P(x,y,z) \rightarrow P(r,\theta,\phi)$. Consider a spherical shell volume $\delta V$ at radius $R_n$:

$$\delta V = \left[ \iint (R_n d\theta)(R_n sin\theta d\phi) \right]\delta r = 4\pi R_n^2 \delta r \tag{A1}$$

Let us say that the shell thickness is equal to the cell size, $\delta r = \Delta$. Points within a shell are denoted by $P_n(R_n, \theta, \phi)$, while flow field variables within a shell are denoted by $\Phi_n(R_n, \theta, \phi)$. Then, one can define a mean flow field variable by

$$\overline{\Phi(R_n, t)} = \frac{1}{\delta V(R_n)} \iiint \Phi(R_n, \theta, \phi, t)dV \cong \frac{1}{N} \sum_1^N \Phi_n \tag{A2}$$

Given the mean $\Phi(R_n, t)$, one can then define the fluctuation by

$$\overline{\Phi'(R_n, t)^2} = \frac{1}{\delta V(R_n)} \iiint \left[ (R_n, \theta, \phi, t) - \overline{\Phi(R_n, t)} \right]^2 dV \cong \frac{1}{N} \sum_1^N \left[ \Phi_n - \overline{\Phi(R_n, t)} \right]^2 \tag{A3}$$

Then the root mean squared (rms) fluctuation becomes

$$\Phi'(R_n, t)_{rms} = \sqrt{\overline{\Phi'(R_n, t)^2}} \tag{A4}$$

The accuracy of the azimuthally averaged variables depends on the ensemble size, $N$. The number of points in the ensemble is shown in Table A1.

**Table A1.** Ensemble size, $N$.

| $R_n$ (cm) | $N = 4\pi(R_n/\Delta)^2$ |
|:---:|:---:|
| 1 | 2000 |
| 5 | $0.5 \times 10^5$ |
| 10 | $2 \times 10^5$ |
| 15 | $4.6 \times 10^5$ |
| 20 | $8 \times 10^5$ |
| for $\Delta_2 = 0.8$ mm. | |

For radii outside of 5 cm, one has $10^5$ samples, which gives a good average. For radii inside 5 cm, there may not be enough points to give an accurate average.

## Appendix B  Helmholtz Velocity Decomposition

The complete velocity field $u = u_\Delta + u_\omega$ is computed by the second order Godunov scheme [16,17]. According to Equation (25), both the dilatational and rotational velocity components scale according to

$$U_\Delta = u_\Delta/a_a = f_\Delta(x, y, z, \tau) \tag{A5}$$

$$U_\omega = u_\omega/a_a = f_\omega(x, y, z, \tau) \tag{A6}$$

In particular, the above points out that the rotational velocity component $u_\omega$—i.e., the turbulent velocity field—also scales with the *explosion length scale*, $r_0$.

Under suitable asymptotic behavior at infinity, an arbitrary velocity vector field $u$ can be decomposed into two parts: a dilatational component $u_\Delta$, which is irrotational, plus a rotational component, $u_\omega$, which is solenoidal:

$$u = u_\Delta + u_\omega \tag{A7}$$

where
$$u_\Delta = \nabla\phi \text{ with } \nabla \times u_\Delta = 0 \tag{A8}$$

$$u_\omega = \nabla \times A \text{ with } \nabla \cdot u_\omega = 0 \tag{A9}$$

with $\phi$ and $A$ denoting scalar and vector potentials, respectively. They satisfy the following Poisson equations:

$$\nabla^2 \phi = \nabla \cdot u \tag{A10}$$

and
$$\nabla^2 A = \nabla \times u \tag{A11}$$

For more details see Batchelor [30], Morino [31] and Hodge [32].

## Appendix C  Nusselt Number Correlation for Turbulent Flow

Gunn [22] has correlated the nondimensional heat transfer coefficient of a sphere, the Nusselt number, with experimental data for water and air. Results are shown in Figure A1. The data are correlated with the following function:

$$Nu = \left(7 - 10e + 5e^2\right)\left(1 + 0.7Pr^{1/3}Re^{1/5}\right) + \left(1.33 - 2.4e + 1.2e^2\right)Pr^{1/3}Re^{0.7} \tag{A12}$$

where $e$ denotes the bed voidage, $Re$ represents the Reynolds number, and $Pr$ is the Prandtl number. For a voidage of 1, this reduces to the following:

$$Nu = 2 + 1.4Pr^{1/3}Re^{1/5} + 0.13Pr^{1/3}Re^{0.7}\,(for\ e = 1) \tag{A13}$$

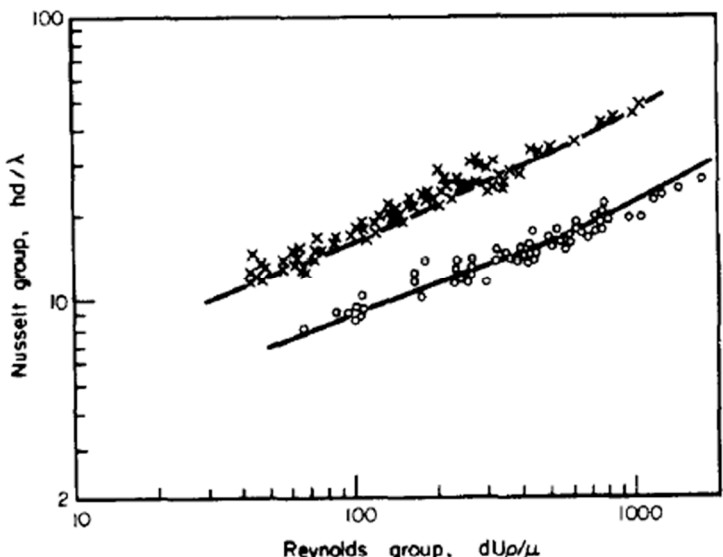

**Figure A1.** Experimental results of Rowe, Claxton, and Lewis for heat transfer to a single sphere comparted with full lines calculated from equation (C1); X water, O air.

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
