# Peer review of "Scaling Turbulent Combustion Fields in Explosions"

_applsci, doi:10.3390/app10238577_

Round 1

Reviewer 1 Report

The paper needs some formatting option (eg equations).

The overall significance of the topic covered by the author is very specific but deserves attention.

The introduction should be re-written with more clarity and simplifications. It is really hard to follow the paper with several known data, old experiments and new calculations.

E.g.: do the reader that is interested in these topics really need the conservation equations and the several standard equations? 

Or: the pictures Fig 1 and Fig 17 seem the same. Please be aware of copyright issues.

Provided these issues the paper can be accepted.

Author Response

  1. Paper Needs Formatting: Corrected formatting issues with equations and References
  2. re-right INTRODUCTION: Introduction has been expanded to include a historical background, starting in 1922......
  3. Do you need to write out Conservation Laws: to establish Similarity of Sedov, one must specify the partial differential equations, initial conditions, equations of state and non-dimensional constants (Reynolds number, Peclet number, Nussult number, etc....) This is what we have done to illustrate the problem as formulated, satisfies Sedov similitude conditions.
  4. In Figure 1, the MAUI/LUX simulation (Fig. 1b) is compared to a field test (Fig. 1 a). In Figure 17, the MAUI/LUX simulation (Fig. 17c) is compared to a vorticity cross-section (Fig. 17a) and a fireball photograph (Fig. 17 b). Both sets of comparisons are necessary
  5. Revisions have been implemented as requested by Reviewer 1

Reviewer 2 Report

The paper considers an interesting fundamental topic of turbulent flame scaling, approached by numerical modeling and processing of the results according to the known similarity laws (the exprosion length scale concept). The results are valuable and cover not only purely gaseous combustion, but also 3-phase pyrotechnic explosions containing dispersed phase for which the applicability of scaling laws must be prooved.

Generally, the paper is well-written, the computational results are obtained by the method being developed by the authors over years (adaptive mesh refinement, gas dynamics solver based on the unsplit high-order Godunov scheme), the conclusions are sound and supported by the results.

However, several issued must be refined to improve the paper presentation quality:

  1. The Introduction section is too concise, it contains no review of relevant research and looks more like Problem Formulation. The paper would benefit from at least short overview of spherical scale and explosion similarity research.
  2. In Eqs. (1)-(5), the species mass fractions are Y_D and Y_A, whereas in the following equations (8)-(11), the mass fractions considered are Y_F, Y_O, and Y_P. How these two sets of mass fractions are related? Both Y_D and Y_F are referred to as "detonation products" - do they coincide or denote different quantities?
  3. By summing up equations (8) to (10), one obtains a transport equation for the sum Y_F+Y_O+Y_P containing no source term. If Y_F = 1 in the fuel, and Y_O = 1 in the oxidizer, then this sum is constant throughout the solution. Where the conservation equation (11) Y_F + Y_O = Y_P follows from? 
  4. In Eq. (12) Δρ_A and Δρ_F are not explained (and not included in the Notation)
  5. Why use such uncommon notation for time as μs/g1/3 or radius cm/g1/3 when proper non-dimensional time and coordinates were introduced? Perhaps, it would be easier for reader to see the time and radius in dimensional and non-dimensional form given together?
  6. On page 21, the bullet points state that the full-scale charge possess the mass of 7,175 kg, whereas the 1/3 scale-charge has the mass of 155.1 kg. A simple calculation gives 7,175/33 = 265.7 kg. Is there any reason for this discrepancy? It must be clarified in the text.

Author Response

  1. Introduction is too concise. Introduction has been expanded to provide a historical perspective dating back to 1922.
  2. Questions about Eqs. (4-5) and( 9-12). Equations (4) & (5) are used are used as arguments in the Equation of State specification {Equations 6 & &). Equations (8 -12) are needed to specify the combustion rate.
  3. see above answer
  4. Rho-air and Rho-F added to Notation
  5. The notation of mu/g^1/3 and cm/g^1/3 are short hand notation used commonly in literature on the subject. See reference [30] Brode (1959)
  6. Error corrected to read "265.7 kg"